# Gold Nanoparticles Prepared with Cyclodextrin Applied to Rapid Vertical Flow Technology for the Detection of Brucellosis

**DOI:** 10.3390/bios12070531

**Published:** 2022-07-15

**Authors:** Ashe Fang, Danni Feng, Xiushuang Luo, Feng Shi

**Affiliations:** College of Life Sciences, Shihezi University, Shihezi 832003, China; fangashe@163.com (A.F.); fdanny@163.com (D.F.); luoxiushuang7@163.com (X.L.)

**Keywords:** brucellosis, gold nanoparticles, signal multiplication, rapid vertical flow technology

## Abstract

Currently, brucellosis seriously threatens the health of humans and animals and hinders the development of animal husbandry. However, the diagnostic methods for brucellosis have some disadvantages, such as low sensitivity, long detection time, professional operation, and high cost. This study aims to establish a convenient, fast, effective, and inexpensive detection method for brucellosis. Gold nanoparticles with β-cyclodextrin as a reducing agent were prepared and optimized, applied to rapid vertical flow technology (RVFT), and used to establish a kit for the detection of brucellosis. In this study, gold nanoparticles prepared from β-cyclodextrin were applied to RVFT for the first time, and on this basis, silver staining amplification technology was introduced, which further improved the sensitivity and reduced the detection limit of this method. Standard Brucella-Positive Serum (containing Brucella antibody at 4000 IU/mL) could be detected in this system even for a dilution factor of 1 × 10^−3^. The detection limit was 4 IU/mL. RVFT is simple to operate, has a short reaction time, and is 5–6 min visible to the naked eye, without any equipment.

## 1. Introduction

Brucellosis seriously endangers human health and the rapid development of animal husbandry. It is a zoonotic disease of animal origin and natural focus. At present, the regions with the highest incidences of this disease in the world are Africa, Asia, Latin America, and the Middle East. Brucellosis is widely distributed all over the world, with brucellosis outbreaks occurring in more than 170 countries and regions. The OIE (Office International Des Epizooties) has listed brucellosis as a statutory animal infectious disease, and China has listed brucellosis as a second-class animal disease [1].

The diagnostic methods for brucellosis include PCR technology, nucleic acid probe technology, loop-mediated isothermal amplification technology, serum agglutination tests, complement fixation tests, enzyme-linked immunosorbent assays, gold nanoparticle-labeled immunochromatography, and so on.

PCR technology is a specific method of detection performed on peripheral blood and other tissues [2], and standard PCR is simple and effective [3]. However, although the traditional PCR methods for detecting *Brucella* are simple and effective, they must rely on specific primers. The detection of different strains of *Brucella* depends on different primers, and only some of them are used for human samples. PCR has been successfully used to identify all *Brucella* species and most biotypes, improving upon the capabilities of traditional molecular genotyping [4,5]. Bruce-ladder multiplex PCR can distinguish all classical *Brucella* species, including those isolated from marine mammals, S19 and RB51 strains, and Rev.1 vaccine strains [6]. PCR is valuable as a tool for detecting *Brucella*. PCR, a diagnostic method for detecting human brucellosis, has the advantages of safety, sensitivity, and specificity [7]. This technology should be considered to complement or confirm the traditional methods, i.e., serological methods or bacterial isolation and culture methods [8]. Due to the shortcomings of this method, such as high cost, quality control, and quality assurance, it is necessary to further evaluate the feasibility of PCR in the routine laboratory testing of clinical samples [9,10].

Mater et al. [11] prepared digoxigenin-labeled probes to identify *Brucella*. In 2000, Fernandez et al. [12] used the 16S rDNA of *Brucella* bovis as probes to detect all *Brucella* strains, varieties, and nine clinical isolates.

Loop-mediated isothermal amplification (LAMP) is a new type of nucleic acid amplification technology that has the advantages of low requirements on experimental instruments and simple operation in the detection process [13]. Mishra Adarsh et al. designed two sets of LAMP primers for the omp2b gene of *Brucella* and standardized detection for all main *Brucella* strains. The results showed that the relative sensitivity of the first primer group was 0.34 pg, that of the second primer group was 34 fg, and that of the bcsp31 PCR was 3.4 pg [14]. Lin Guozhen et al. [15] established a new LAMP method for detecting *Brucella* DNA in animal blood, human blood, and dairy products.

The serum agglutination test is one of the most widely used methods for the diagnosis of brucellosis. It is a standard and highly sensitive method [16]. Traditional serum agglutination tests of *Brucella* include the standard tube agglutination test (SAT), plate agglutination test, milk ring test (MRT), and anti-human immunoglobulin test (Coomb′s). In one study, the sensitivity of an enzyme-linked immunosorbent assay (ELISA) was 81.3% and that of an SAT was 93.7% [16]. Many studies have confirmed that the sensitivity of SAT can reach 100% [17]. Although the sensitivity of the serum agglutination test is very high, there are some limitations with regard to false-positive or false-negative results [16,18,19].

Compared with the SAT and the Rose-Bengal plate agglutination test (RBPT), the complement fixation test (CFT) has a higher specificity and sensitivity [20]. The Office International des Epizootics (OIE) has recognized that the CFT is a definitive test for the determination of brucellosis and has high application value in clinical diagnosis. However, the CFT also has shortcomings. Its detection technology is mainly applied for the diagnosis of brucellosis in cattle, sheep, and sheep epididymis species and is not suitable for the detection of a variety of brucellosis types, especially for the individual diagnosis of brucellosis in pig species. After an experimental study, the sensitivity was reduced, and samples with hemolysis could not be detected with the CFT because the hemolysis samples had more components participating in the reaction during the experiment, and multiple groups of controls would be required. In addition, the interpretation of the results required visual observation, and the hemolysis samples would interfere with the subjective interpretation of color depth, so the subjective influence was large in the detection of the hemolysis samples, and errors were easily generated. In addition, the CFT had high experimental requirements and complicated operations, so it was difficult to apply to grassroots quarantine.

Enzyme-linked immunosorbent assay (ELISA) can be divided into two types: indirect ELISA (iELISA) and competitive ELISA (cELISA). For the detection of brucellosis, ELISA can be used with samples of not only serum but also milk. Kittelberger pointed out that iELISA has high sensitivity, which is superior to those of the other serological methods mentioned above [21], but in the actual diagnosis of brucellosis, it is still difficult to distinguish artificial immunity from natural infection, and there is still the problem of serum crossover. To solve these problems in the diagnosis of brucellosis, Nielsen [22] created the cELISA method. The sensitivity of cELISA was similar to that of ELISA, but it was more sensitive than that of the buffered plate agglutination test (BPAT). cELISA has greater specificity than iELISA when detecting false-positive samples or animals not immunized with the *Brucella* vaccine. The sensitivity of CFT has been reported to be lower in sheep in field conditions (88.6%) than those of RBT (92.1%) and iELISA (100%) [23,24]. Therefore, in 2004, the OIE designated cELISA as the recommended method for the diagnosis and elimination of bovine brucellosis.

RVFT was originally established by Spielberg et al. in 1989 for the detection of antibodies to human immunodeficiency virus [25]. RVFT uses NC film as the reaction carrier, and uses the filterability of microporous membrane to complete the immune reaction and washing on a special diafiltration device through liquid diafiltration through the membrane. This process has an affinity chromatography. Concentration can achieve the purpose of rapid detection. Using gold nanoparticles as a chromogenic marker makes the positive results appear as red spots on the carrier film, and the detection results can be qualitatively judged with the naked eye. The detection process is easy to operate and does not require any auxiliary equipment, and is widely used in food testing, clinical disease testing, animal disease testing, and other fields [26,27,28].

The main advantages of RVFT are as follows:

(1) RVFT is easy to use and simple to operate, and users do not need special training. (2) RVFT is safe, non-toxic, and has no side effects. As a gold nanoparticle itself has color, it does not need to have added color developer and stop solution, and avoids the use of radioisotopes and toxic substrates/organisms. (3) RVFT is economical and environmentally friendly: the sample basically does not need to be pretreated, and can be tissue fluid, serum, and urine, eliminating the need for sample pretreatment in traditional analysis methods. (4) RVFT detection time is short: RVFT can generally obtain the detection result in 2~3 min, which is unattainable by other detection methods at present. (5) The detection cartridge is small in size and easy to carry: RVFT′s detection cartridge is suitable for field operations or on-site temporary detection, and is not limited by the experimental site and conditions. (6) RVFT production cost and detection cost are both low: the number of reagents and samples is small, and the sample volume can be as low as 10 μL, which reduces the cost of detection; at the same time, the materials for preparing the detection cartridge are cheap, large-scale equipment is not required, and the production cost is very low. (7) The experimental results can be stored for a long time: the prepared strip can be stored for one year or even longer in the refrigerator at 4 °C. (8) There are many types of test specimens: RVFT can be used to test tissue fluid, urine, or feces, etc., so it is suitable for various tests. (9) RVFT has a wide range of applications and can be used for detection in many clinical and non-clinical fields.

The main disadvantages of RVFT are as follows:

(1) Compared with the lateral flow immunochromatography technique, the method is relatively simple. For example, lateral flow immunochromatography techniques include competition method, sandwich method, indirect method, and so on. Therefore, the substances that can be detected are limited. (2) This technique requires additional synthesis buffer and chromogenic solution.

In traditional RVFT, classic trisodium citrate is mostly used to prepare gold nanoparticles, which are marked and made into a detection card box. In this study, spherical gold nanoparticles prepared using cyclodextrin had strong stability, and as a labeling material, a detection cassette of red gold nanoparticles was prepared.

Cyclodextrins are ideal host molecules that are the most similar to enzymes found thus far, and they have the characteristics of enzyme models. Therefore, cyclodextrins have received great attention and have been widely used in the fields of catalysis, separation, food, and medicine. Due to the solubility and inclusion ability of cyclodextrins in water, changing the physical and chemical properties of cyclodextrins has become one of the important purposes of their chemical modification.

In the pharmaceutical industry, cyclodextrin can effectively increase the solubility and dissolution rate of some drugs with poor water solubility in water and improve drug stability and bioavailability. Since β-cyclodextrin is composed of glucose, it is non-toxic, harmless, free of side effects, and can be absorbed by the human body [29]. Therefore, it has received much attention in the pharmaceutical industry. It is widely used as a filler and binder in medicine. It has the general properties of starch, and the prepared clathrate is easy to powder, can be made into powder and tablets, is convenient to store, and has good dosage forms. The application of β-cyclodextrin in the medical industry has wide and attractive prospects. In addition to the pharmaceutical industry, β-cyclodextrin is also widely used in the food industry and the daily chemical industry, as well as in analytical chemistry, environmental protection, and cosmetics.

Cyclodextrin is a cyclic oligomer obtained from starch by enzymatic degradation which was discovered by French pharmacist Villiers in 1891. Cyclodextrins have remarkable capability to establish supramolecular host–guest interactions because of their toroidal shape and non-polar inside [30,31]. The less-hydrophilic internal cavity and the hydrophilic exterior of β-CD enable it to make supramolecular interactions such as electrostatic, hydrogen bonding, van der Waals forces, and hydrophobic interactions with organic, inorganic, and bio molecules [32]. Cyclodextrin molecules contribute distinguished advantages due to their novel architectural features to form inclusion complexes with several kinds of molecules such as ions, protein, and oligonucleotides [33].

In this experiment, cyclodextrin functionalized gold nanoparticles were synthesized. As the only reducing agent and stabilizer, cyclodextrin modified gold nanoparticles were synthesized by one-step green method without adding other reducing agents and stabilizer. This method not only provides a new direction for chemical synthesis, but also provides a diversified direction for the application of gold nanoparticles.

Gold-labeled silver staining amplification technology has been well-applied in chiral amino acid detection [34], electrochemical immunoassays [35], and DNA analysis [36] because of its simple operation and obvious amplification of response signals. The basic principle of gold-labeled silver staining amplification technology is that silver atoms are deposited around gold nanoparticles by the catalytic reduction reaction of silver ions to form a silver shell, thus realizing the target. The silver deposition strategy has the advantages of strong signal amplification and simple operation, and has great potential in practical applications.

In this experiment, gold nanoparticles prepared by β-cyclodextrin were used as labeling materials, SPA was labeled, and red RVFT was established. On this basis, a gold-labeled silver staining technique was performed to further amplify the detection signals, improve the sensitivity, and reduce the detection limit, which provided theoretical support for the establishment of a convenient, efficient, sensitive, and low-cost detection method for brucellosis.

## 2. Materials and Methods

### 2.1. Experimental Reagents

HAuCl_4_·3H_2_O, K_2_CO_3_, Immunoglobulin (IgG), PVP-40, BSA, NaCl, Tween-20, EDTA, NaN_3_, Sucrose, Tris, β-cyclodextrin, and staphylococcal protein A (SPA) were purchased from Sigma (St. Louis, MO, USA). A lipopolysaccharide extraction and purification kit was purchased from Intron Biotechnology Company (Chengnan, Korea). The water used in this study was deionized. The cover, detection box, and nitrocellulose filter membrane (NC) were all purchased from Advanced Microdevices (Haryana, India).

### 2.2. Laboratory Apparatus

The laboratory apparatus used in this study include a Molecular Devices microplate reader, a Malvern particle size analyzer, an ultrasonic cleaning instrument, an ultra-pure water instrument, a ten-thousandth analytical balance, an electrothermal constant temperature drying oven, a freezing centrifuge, and a magnetic stirrer.

### 2.3. Preparation of Gold Nanoparticles

At 200 °C, 1% chloroauric acid solution was added to 50 mL ultra-pure water at 600 rpm and stirred for 5 min. Then, β-cyclodextrin (1% *w/v*) and 120 μL NaOH (1 M) were quickly added. When the solution turned wine red, heating stopped and stirring occurred for another 10 min. There was a color change from colorless-light to pink–purple–burgundy.

### 2.4. Marker Protein A

The prepared gold nanoparticle solution was filtered by using a 0.22 µm aqueous phase needle filter, and then the pH of the gold nanoparticle solution was adjusted to approximately 6.5 by using 0.2 mol/L K_2_CO_3_. Under uniform stirring, 1 mg/mL *Staphylococcus aureus* protein A was slowly added. The stirring continued for 30 min. The container was allowed to stand for more than 10 h under sealed conditions so that the gold nanoparticles and *staphylococcus aureus* protein A were fully coupled. After standing, BSA was added, and the mixture was allowed to stand for more than 1 h to obtain a blocking effect. The mixture was then transferred to a centrifuge tube using a pipette and centrifuged at 8000 rpm for 30 min under constant weight. After centrifugation, the supernatant was slowly discarded with a pipette gun, the complex solution (the complex solution consisted of 1 mL of 5% bovine serum albumin, 500 μL of 5% sucrose, 500 μL of 5% trehalose, 1 mL of 10% polyvinylpyrrolidone, and 7 mL of Tris HCl, pH 8) was added, and the mixture was repeatedly blown, aspirated, and mixed. Ultrasound was performed for 2 min with an ultrasonic instrument, and the gold nanoparticles were suspended in the solution for subsequent use at 4 °C.

### 2.5. Preparation of Silver Dye Reinforcing Solution

Solutions of silver nitrate (0.3% *w*/*v* aqueous solution) and hydroquinone (3% *w*/*v* aqueous solution, 0.5 M citric acid buffer, pH 4.0) were prepared and stored at room temperature in dark conditions. This solution was added and incubated for 30–60 s to observe the results.

### 2.6. Assembly of Rapid Vertical Flow Technology Device

The RVFT device comprises a comprehensive buffer solution, a developing solution, and a detection card, wherein the comprehensive buffer solution facilitates the functions of diluting the sample, wetting the NC film, and washing the background. The color-developing solution is a complex solution after gold nanoparticle-labeled *Staphylococcus aureus* protein is centrifuged; the detection card comprises a lower cover body, a nitrocellulose membrane, water-absorbing paper, and an upper cover body. The NC film was cut into a rectangular shape with a length of 1.8 cm and a width of 1.2 cm by a cutting machine, and absorbent paper was cut into a rectangular shape with a length of 2.5 cm and a width of 1.8 cm. In this experiment, silver staining amplification technology was additionally introduced, so the cartridge device was required to have a strong water absorption capacity. Based on the original RVFT, filter paper with a better water absorption capacity was incorporated. Three layers of absorbent paper were placed at the bottom of the detection box, and the NC film was placed on top of the absorbent paper (as shown in Figure 1).

### 2.7. Selection of NC Film

Different types of NC membranes with different pore diameters and with or without backing were treated with purified lipopolysaccharides (LPS) for the diafiltration experiment, and the reaction background uniformity, color, and diafiltration speed were observed. According to the corresponding experimental results, an NC membrane with a moderate infiltration speed, uniform background, and light color after the reaction was selected for this experimental study.

### 2.8. Optimization of Coating Antigen

In RVFTs, coated antigen is very important because it directly affects the whole reaction system and the sensitivity of the experimental results. *Brucella* have no spore, capsule, flagellum, or other structures, and therefore most of them lack motility. *Brucella* is divided into smooth and rough. *Brucella* is mostly smooth-type without LPS-O side chains. The contents of G and C in *Brucella* nucleic acid were relatively high, accounting for 55–58% of the total [37]. Polysaccharide O antigen, polysaccharide nucleus, and lipid A are the three main components of LPS. LPS is a membrane component of Gram-negative bacteria and plays an important role in maintaining the stability of cell function. *Brucella* LPS is also an endotoxin of *Brucella* and an important virulence factor. Based on the above properties of LPS, LPS was selected as the coating antigen in this study. The quality control point is used to judge whether the test cartridge is valid. In RVFTs, recombinant SPA is used as the labeled protein, because recombinant SPA has high affinity and can bind to mammalian IgG Fc fragments. Therefore, the quality control points can be coated with sheep IgG (Figure 2).

### 2.9. Implementation of the Test Steps and Interpretation of the Test Results

The detection of antibodies involves these steps. First, sample dropping steps: the sample is serum sample or tissue fluid, etc., and the absorbed sample is directly dropped into the visible window of the filtration detection card. After the liquid is completely soaked, the color development step can be performed. Second, the chromogenic steps: add 3~4 drops of colloid gold to the detection card to mark the chromogenic solution prepared by SPA. After the chromogenic solution is fully permeated, drop 3 drops of comprehensive buffer solution to rinse the impurities. Third, observe the test results: after all the liquid infiltration, observe the test results in the observation window of the test card.

Red dots appeared at the positions corresponding to the detection points and the quality control points in the observation window, and the interpretation of the result was positive regardless of the color depth (Figure 3a). If the corresponding red dot appeared at the position corresponding to the control point in the observation window, it was interpreted as negative (as shown in Figure 3b). If the detection point had a red dot, or if there was no red dot, the detection cartridge was interpreted as invalid.

## 3. Results

### 3.1. Optimization of the Synthesis Process of Gold Nanoparticles

The size and shape of the gold nanoparticles can directly affect their coupling effect with SPA and affect the sensitivity of the RVFT device. For example, Schleh et al. [29] found that the size and surface charge of gold nanoparticles determine their usefulness. The quality of gold nanoparticles is directly related to the reaction. Before the reaction, the concentration of reactants is critical. In the process of the reaction, the mixing method, adding sequence, reaction temperature, and so on, as well as the cleaning of the reaction vessel, will affect the quality of the final gold nanoparticles. In this study, the synthesis of gold nanoparticles with β-cyclodextrin was optimized several times. The optimization process is as follows.

First, the optimal reaction temperature for the preparation of gold nanoparticles was determined. According to the actual experimental results, the reaction temperature gradient was set to 100 °C, 150 °C, 200 °C, 250 °C, and 300 °C, and the reaction conditions, such as the ratio of HAuCl_4_ to β-cyclodextrin, reaction speed, and pH value, were maintained as constants by using the controlled variable method to determine the optimal reaction temperature.

UV–visible spectroscopy is a very helpful technique which can be used to estimate size, concentration, and aggregation level of gold nanoparticles. The Localized Surface Plasmon Resonance (LSPR) spectrum depends upon the size and shape of gold nanoparticles. The peak absorbance SPR wavelength increases with particle diameter because the distance between particles decreases due to aggregation [38]. Enhanced SPR peak intensity for smaller nanoparticles (40 nm) compared to larger ones (80 nm) has been reported by Zeng and co-workers [39]. Zuber et al. [38] reported a shift in SPR maximum towards longer wavelength with increase in size of gold nanoparticles due to the differences in the frequency of surface plasmon oscillations of the free electrons. The absorbance was also found to be increased with increase in size of nanoparticles due to the enhanced mean free path of the electrons in the larger nanoparticles [40].

The resulting data, such as the maximum absorption wavelengths, OD values, and peak widths of the UV–Vis spectra, were used to preliminarily judge whether the prepared gold nanoparticles were uniform in size and dispersed at different temperatures (Figure 4).

As we all know, temperature is an important determinant of the catalytic performance for catalysts [41,42]. Therefore, temperature will affect the quality of gold nanoparticles, and different sizes and shapes of gold nanoparticles will be formed at different temperatures. As seen from Figure 4, when the temperature gradually rises from 100 °C to 200 °C, the OD value of the prepared gold nanoparticles gradually increases, and when the temperature continues to rise, the OD value shows a downwards trend. At 200 °C, the peak value of the UV–visible spectrum is 528 nm, with the narrowest peak width and the largest OD value, which indicates that the size of gold nanoparticles is uniform and the dispersion is good, so the optimum temperature for preparing gold nanoparticles by β-cyclodextrin is 200 °C.

The optimal molar ratio of the reaction substances for preparing the gold nanoparticles was determined. According to the actual experimental results, the values of the molar ratios of the reaction substances were selected to be 1:1, 1:2, 1:3, 1:6, and 1:9, and the reaction conditions, such as the reaction temperature, the reaction rotating speed, the pH value, were maintained as controls by using the controlled variable method to determine the optimal molar ratio of the reaction substances.

The ultraviolet–visible light spectral curves of the gold nanoparticles prepared with different molar ratios of reaction substances were measured by using a microplate reader, and the absorbance values and peak widths were compared. The larger the absorbance was, the higher the concentration of the gold nanoparticle solution was, and the narrower the peak width was, indicating that the dispersibility of the gold nanoparticles was better. As seen from Figure 5, the curve data of the 1:6 group were preferable, indicating that the optimal molar ratio of the reaction substances was 1:6.

### 3.2. Characterization of Gold Nanoparticles

After AuCl_4_^-^ is reduced to elemental gold, a plasma resonance ultraviolet absorption characteristic peak of gold nanoparticles appears near 510–550 nm. The position, intensity, and width-at-half-maximum of this absorption peak are closely related to the size, number, and dispersion state of the nanoparticles [34]. The gold nanoparticle solution prepared with β-cyclodextrin as the reducing agent presented a bright wine-red color (Figure 6a), with a uniform color and no delamination phenomenon, floaters, or precipitates. When the solution was irradiated with a laser pointer, a sharp straight-line beam—the Tindal effect—appears, indicating a colloidal state (Figure 6b).

After being left alone for six months, the state of the solution was observed to be unchanged, indicating that the stability was good. After the preparation of gold nanoparticles, with the passage of time, the characterization of transmission electron microscope was carried out every month, as shown in Figure 7, to further observe the dispersion phenomenon of gold nanoparticles and whether there was agglomeration. The results obtained according to Figure 7 show that the gold nanoparticles have good dispersibility and do not show a large amount of agglomeration after being placed for 6 months.

The absorbance of the gold nanoparticle solution prepared by β-cyclodextrin at 300–700 nm was measured by using a Molecular Devices microplate reader, and the UV–Vis absorption spectrum was obtained, as shown in Figure 8. The surface plasma resonance peak appeared at 519 nm, which was consistent with the color of the solution. The peak type was narrow, and the half-peak width was approximately 30 nm, indicating that the prepared gold nanoparticles were uniform in size and had good dispersibility.

A common carbon support film was completely immersed in the prepared gold nanoparticle solution, and after drying treatment, the gold nanoparticles were observed by using a FEITecnai G2-F20 high-resolution field emission transmission electron microscope. Through TEM analysis, in the field of view of the low-power lens, more gold nanoparticles could be observed, which had good dispersibility and no large agglomeration phenomenon. In the field of view of the high-power mirror, the morphology of the gold nanoparticles was observed to be spherical, and no triangular or other forms of gold nanoparticles were present (Figure 9). The particle size was uniform, with a diameter of approximately 20 nm. Further observations of good dispersibility verified that the prepared spherical gold nanoparticles were very stable and could be stored at 4 °C for further experiments.

### 3.3. Optimization of Marker SPA

The addition of marker SPA needs to be optimized during this process. After experimental verification, when the addition amount of SPA was less than 600 µL, as shown in Figure 10a, the color-rendering effect was relatively shallow. When the amount of SPA was higher than 600 µL, as shown in Figure 10b, the color was clear, and the overall effect was good. In line with the principle of cost savings, the optimal amount of labeled SPA was selected as 600 µL. (This time, as a preliminary condition to explore, in order to save the package material, only the package was a point.)

### 3.4. Optimization of Synthetic Buffer

It is necessary to optimize the washing effect of the comprehensive buffer during this process. Before optimization, the detection background was not clean, and a pink interference signal appeared. In this study, Tween-20 [43], a surfactant, was used to break red blood cells, wet and disperse red blood cells, increase hydrophilicity, and reduce the nonspecific adsorption between antibody and nitrocellulose membrane [43]. The background color appearing on the test was weakened by adding Tween-20. (This time, as a preliminary condition to explore, in order to save the package material, only the package was a point).

As shown in Figure 11, when the content of Tween-20 is less than 8%, the interference signal in the background is relatively strong, which will affect the result judgment. When the content of Tween-20 is greater than 8%, the color of gold nanoparticles is weakened and the signal is reduced. Judging from the above situation, the content of Tween-20 is finally determined as 8%.

### 3.5. Detection of the Specificity of the Diafiltration Card

In the specificity evaluation experiment (Figure 12), RVFT was used to determine *Brucella*-positive serum from sheep serum from mice, *Enterococcus*-positive serum from sheep, *Mycobacterium tuberculosis*-positive serum from humans, *Escherichia coli*-positive serum from rabbits, and *Staphylococcus aureus*-positive serum. However, only the *Brucella*-positive serum was positive, and all other serum samples were negative, suggesting that no cross-reactivity or specific reactions occurred.

### 3.6. Detection of the Detection Limit of the Percolating Card

The test results of the standard *Brucella*-positive serum (containing *Brucella* antibody at 4000 IU/mL) diluted with normal saline at concentrations of 1:10, 1:50, 1:100, 1:1000, and 1:10,000 are shown in Figure 13. As the serum concentration decreases, the color of the T-point becomes lighter. Visual observations were still possible with the red cassette at a dilution of 1:100, demonstrating the high sensitivity of RVFT for detecting low antibody levels.

### 3.7. Determination of Optimal Silver Staining Incubation Time

The principle of the technique is to label SPA with gold nanoparticles prepared by cyclodextrin and then introduce silver ions into the system. The small size effect of gold nanoparticles particles causes it to have a strong catalytic reduction ability, which can reduce silver ions in the surrounding system to silver particles. These silver particles can further catalyze the reduction of silver ions in the system. This cascade-like catalytic action makes the silver ions gather more and more, tightly wrap the gold nanoparticles, and finally accumulate into agglomerated silver shells, forming black particles visible to the naked eye.

Reaction time of the process of silver staining was optimized here, as Figure 14 clarifies. The incubation time during silver dyeing was set as a gradient, divided into 1 min, 2 min, 3 min, 4 min, and 6 min. Experimental results show that when the incubation time is less than 3 min, the time was too short to transform enough silver ions into silver metal, leading to inconspicuous color change of the strip. However, when the reaction time increased to 6 min, metallic silver produced by the process was enormous enough to be detected outside the T-point and C-point on the membrane. The suitable incubation time was 3 min. Gold nanoparticles could be fully covered by silver ions and resulted in distinct black color visualized on the T-point and C-point without diffusion.

### 3.8. Silver Staining Results

As shown in Figure 15, the signal intensity at the T-point gradually decreased with decreasing positive serum concentration. In the absence of silver deposition (Figure 15), only very weak red dots were obtained from the T-point as the serum concentration decreased, and when diluted to 1:100, the T-point was equivocal. After amplification with silver staining, the serum dilution ratio of 1:1000 could still be clearly observed. After the introduction of the amplification technology of “gold-labeled silver staining” (Figure 15), the color of the T-point changed from red to black, and the T-point, which originally developed a very light color, also became easy to observe. The basic principle of “gold-labeled silver staining” amplification technology is to use the catalytic reduction reaction of gold nanoparticles on silver ions to deposit silver atoms around the gold nanoparticles to form a layer of silver shell so that the color of the color-developing point changes from red to black and the visibility of the color-developing is deepened, thus further improving the sensitivity of RVFT and reducing the detection limit.

### 3.9. Repeatability and Stability Evaluation of RVFT

Stability test: Take three batches of test cartridges, named Cartridge A, Cartridge B, and Cartridge C, and test the standard positive serum for brucellosis three times at a time interval of 15 days.

Repeatability detection: Take cassette I, cassette II, and cassette III to detect brucellosis standard positive serum, and evaluate the repeatability of RVFT according to the experimental results.

The test results are shown in Table 1. The results show that the RVFT has good stability and good repeatability.

### 3.10. The Advantages of RVFT Compared with Lateral Flow Immunoassay

In the current point-of-care detection technology, lateral flow immunochromatography is a commonly used detection method. As reported by Chandan Prakash et al., the lateral flow assay strip test was able to detect 107 cfu/mL B. abortus S99 inactivated organism in PBS [44]. Compared with lateral flow immunochromatography, RVFT has three major advantages. First, RVFT shortens the detection time. The detection time of lateral flow immunochromatography is generally 10 to 15 min. However, the RVFT detection time is 5–6 min, which greatly shortens the detection time. Second, RVFT effectively avoids the hook effect [45], which is an interference associated with excess antigens that do not bind to the conjugated antibody.

## 4. Conclusions

In conclusion, we report an RVFT device for the detection of antibodies to diagnose brucellosis. In this study, spherical gold nanoparticles prepared with β-cyclodextrin were used for the first time to diagnose brucellosis. To improve the sensitivity, silver staining amplification technology was introduced in the later stages to ensure that the technology had the characteristics of high sensitivity, fast detection speed, low cost, simplified operation steps, and no need for sample pretreatment.

β-Cyclodextrin has high enzyme activity. The prepared gold nanoparticles also have certain enzyme activities. It is anticipated that this chemical property can be fully utilized to further enhance the sensitivity and reduce the detection limit by enzyme activity coloration.

With the development of RVFT, an increasing number of gold nanoparticle types prepared with different reducing agents can be used as labeling materials, and it is anticipated that the sensitivity can be improved more effectively through optimizing the source of the materials. With the continuous development of RVFT, it is anticipated that this technology can be applied to more fields to bring increasing convenience to people′s lives and improve their quality of life.

## Figures and Tables

**Figure 1 biosensors-12-00531-f001:**
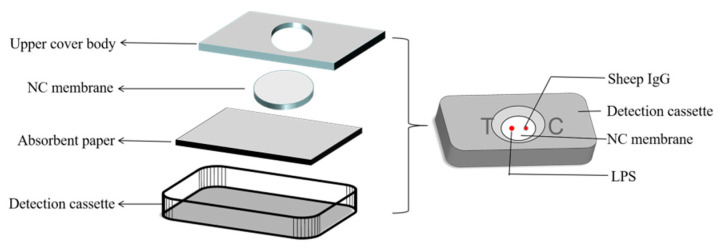
Structure diagram of the test box.

**Figure 2 biosensors-12-00531-f002:**
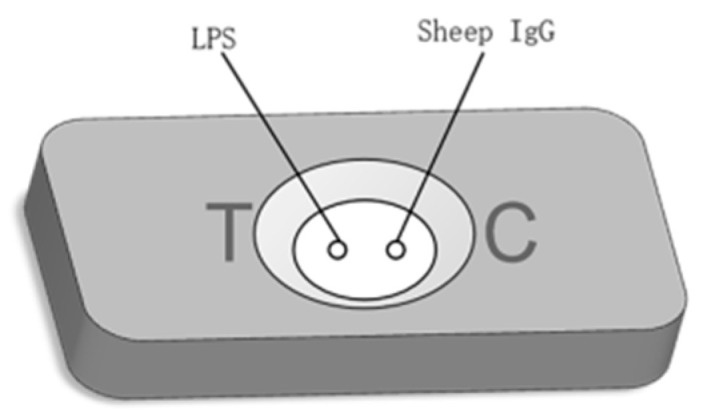
T-point and C-point coating diagram.

**Figure 3 biosensors-12-00531-f003:**
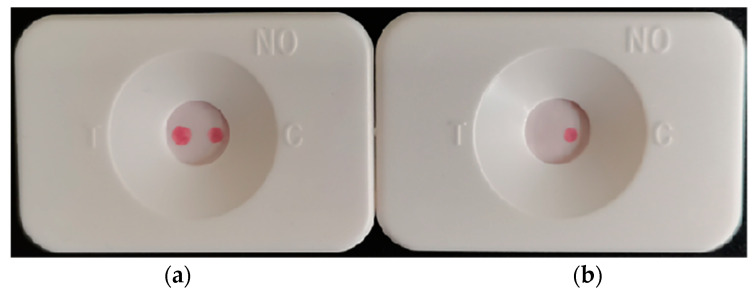
(**a**) Positive result (left); (**b**) Negative result (right).

**Figure 4 biosensors-12-00531-f004:**
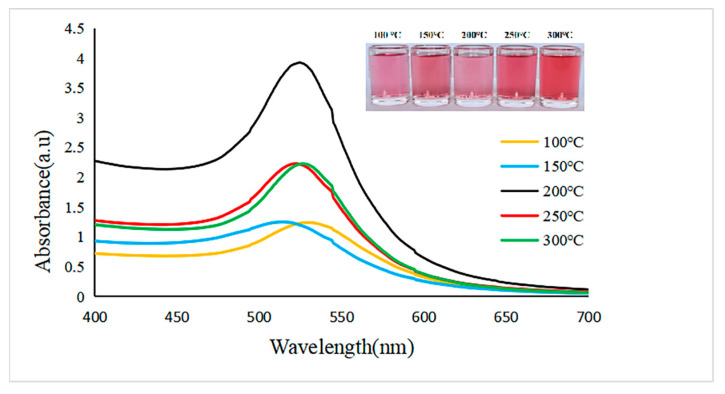
Ultraviolet–visible absorption spectra and colloid color diagram of gold nanoparticles prepared at different reaction temperatures.

**Figure 5 biosensors-12-00531-f005:**
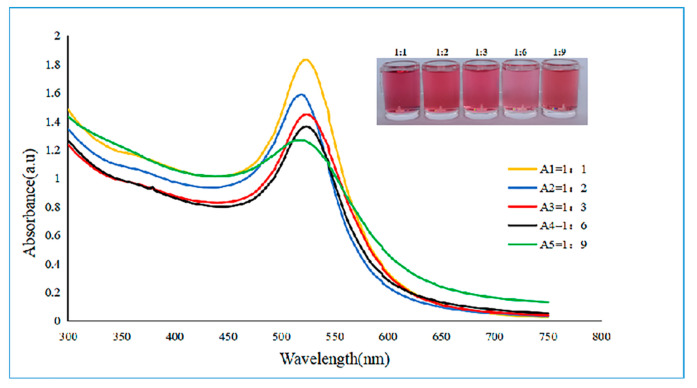
Ultraviolet–visible absorption spectra and colloid color diagram of gold nanoparticles prepared by different molar ratios of reactive substances.

**Figure 6 biosensors-12-00531-f006:**
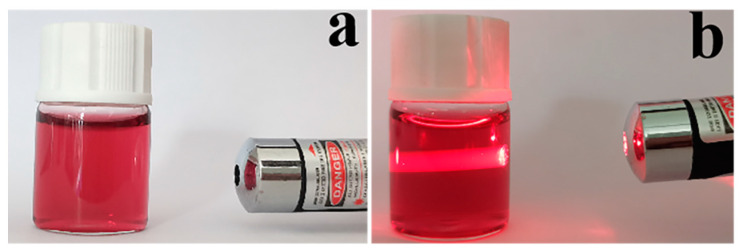
(**a**) Image of the gold nanoparticle solution; (**b**) Tindal effect diagram.

**Figure 7 biosensors-12-00531-f007:**
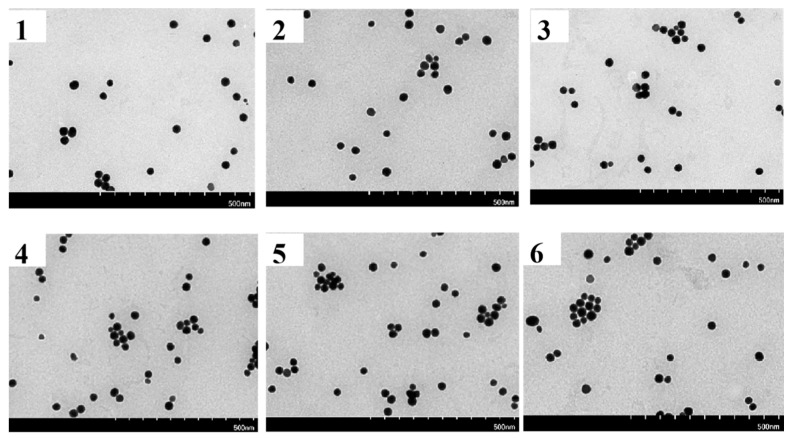
Electron micrographs of gold nanoparticle solutions from 1 to 6 months.

**Figure 8 biosensors-12-00531-f008:**
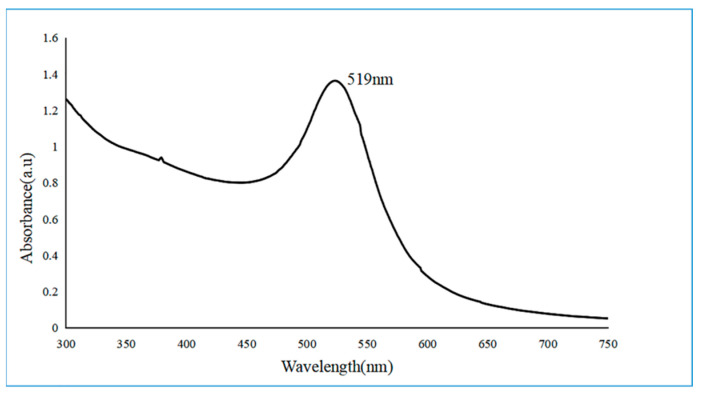
Ultraviolet–visible absorption spectra of gold nanoparticles.

**Figure 9 biosensors-12-00531-f009:**
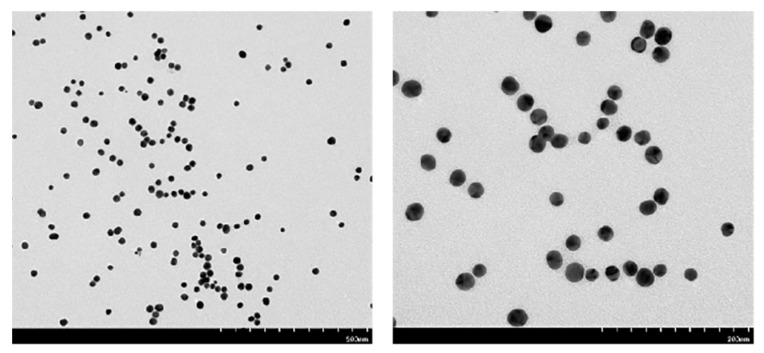
Transmission electron microscopy image of the spherical gold nanoparticles.

**Figure 10 biosensors-12-00531-f010:**
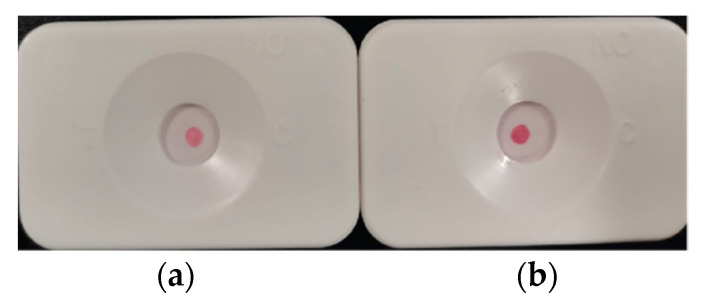
Optimization of SPA content: (**a**) Before optimization; (**b**) Post-optimization.

**Figure 11 biosensors-12-00531-f011:**
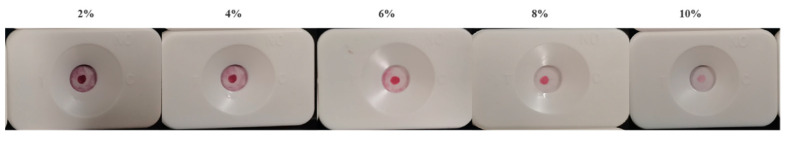
The result diagram of optimizing the content of Tween-20 comprehensive buffer.

**Figure 12 biosensors-12-00531-f012:**
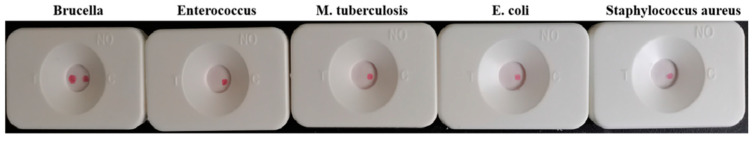
Specific results for RVFT.

**Figure 13 biosensors-12-00531-f013:**
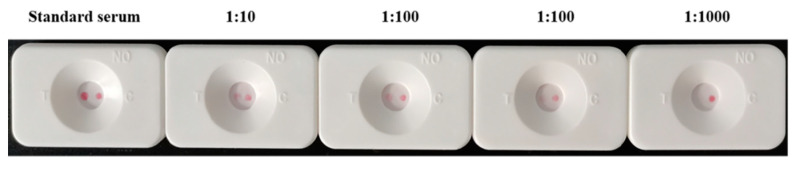
Experimental results of RVFT detection limit.

**Figure 14 biosensors-12-00531-f014:**
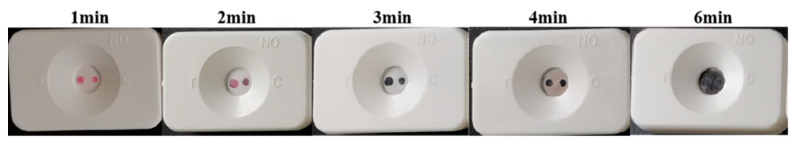
Plot of incubation time results for silver staining.

**Figure 15 biosensors-12-00531-f015:**
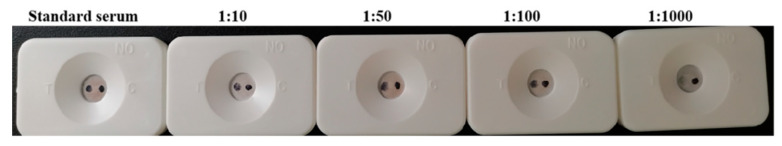
Silver staining results.

**Table 1 biosensors-12-00531-t001:** Test Results of Stability and Repeatability of Percolation Card.

	Standard Brucellosis Positive Serum				Test Card Box		
	1	2	3		I	II	III
Test card box	+	+	+	standard brucellosis positive serum	+	+	+
Test card box	+	+	+	standard brucellosis positive serum	+	+	+
Test card box	+	+	+	standard brucellosis positive serum	+	+	+

## Data Availability

Not applicable.

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
