# Peer review of "Gold Nanoparticles Prepared with Cyclodextrin Applied to Rapid Vertical Flow Technology for the Detection of Brucellosis"

_biosensors, 2022, doi:10.3390/bios12070531_

Round 1

Reviewer 1 Report

Comments

The development or improvement of innovative and sensitive, friendly users and non-expensive methodology to detect pathogens. Nevertheless, the authors present an improvement in the synthesis and application of gold nanoparticles to rapidly detect brucellosis.

The introduction section reports the state of the art and advantages of the methodology with respect to the state of the art. It could be improved by adding additional methods that apply other plasmonic nanoparticles for the detection of biomarkers or pathogens such as Heatsens technology already in the market, or simply which is the main advantage compared with the commonly used lateral flow technology.

Then, the methodology describes the synthesis and characterization of the beta-CD functionalized of gold nanoparticles, as well as the RVTF setup. Despite the UV-VIS and TEM characterization it would be good to report the difference in dimension of nacked Au-nanoparticles and after b-CD. Moreover, the Authors claims that the b-CD is used as reducing agent, but it is not reported any experiments that helps to prove the claim. 

In addition, please report the concentration of b-CD and NaOH (line204)

The results are clearly presented and supported by the figures that show the setup and the results on selectivity. 

The quality of the figures could be slightly improved. On the other hand, the figures’ caption well describe the figure panel. 

The conclusion is in agreement with the results reported.

More in general the paper is well written and easy to read.

I’m glad to consider this manuscript for publication with minor revision.

Author Response

Response to Reviewer 1 Comments

Dear Editors and Reviewers:

Thank you for your letter and for the reviewers’ comments concerning our manuscript entitled

“Gold nanoparticles prepared with cyclodextrin applied to RVFT for the detection of brucellosis”

(Manuscript ID: biosensors-1782721). Those comments are all valuable and very helpful for revising

and improving our paper, as well as the important guiding significance to our researches. We have

studied comments carefully and have made correction which we hope meet with approval. The

modified part of the article has been improved by using the "Track Change" tool. The main

corrections in the paper and the responds to the academic Editor’s comments are as following:

Point 1: Moreover, the Authors claims that the β-CD is used as reducing agent, but it is not reported

any experiments that helps to prove the claim.

Response 1: Thank you very much for your valuable suggestions. In "Three-in-One: Sensing,

Self-Assembly and Cascade Catalysis of Cyclodextrin Modified Gold Nanoparticles", it is explicitly

written that β-CD is used as the reducing agent. The details are as follows:

“ Monodisperse AuNPs with 15-20 nm in diameter are eco-friendly fabricated by the proposed

one-step colloidal synthesis method using CD as both reducing agents and stabilizers.”

[1]Zhao, Y.; Huang, Y.; Zhu, H.; Zhu, Q., (2016). Three-in-one: sensing, self-assembly, and cascade

catalysis of cyclodextrin modified gold nanoparticles. Journal of the American Chemical Society,

138(51), 16645.

Point 2: In addition, please report the concentration of β-CD and NaOH (line204)

Response 2: Thank you very much for your valuable suggestions. We have made dependent

additions in the article.

Thank you for your affirmation of our article and previous revision work, which is greatly

encouraged. At the same time, thank you for agreeing to the publication of the article, which is

undoubtedly the best affirmation of our long-term work. Thank you again!

Thanks again!

We tried our best to improve the manuscript and made some changes in the manuscript. These

changes will not influence the content and framework of the paper. We used the "Track Changes"

function to make corresponding Changes, and the Changes are clearly visible.

We appreciate for Editors/Reviewers’ warm work earnestly, and hope that the correction will meet with

approval.

Once again, thank you very much for your comments and suggestions.

Sincerely yours

Feng Shi

Reviewer 2 Report

Dear authors,

It is a big and extensive work, however I regret to say that I cannot recommed the acceptance of your paper, at least in its current state. My comments/suggestions/concerns are the following:

Abstract:

1. I believe it should be concise. It’s not necessary to detail the optimization of gold nanoparticles here; the sentence “The optimal conditions for preparing the gold nanoparticles were as follows: the optimal reaction temperature was 200 °C, the optimal molar ratio of reactants was 1:6, the maximum appropriate amount of labelled staphylococcal protein A (SPA) was 600 µL, and the maximum appropriate amount of blocking solution was 4 mL. The maximum concentration of Tween-20 in the comprehensive buffer was 8%, and the A solution: B solution ratio in the silver staining reagent was 1:3” can be removed.

Introduction

2. “The diagnostic methods for brucellosis include PCR technology, nucleic acid probe technology, loop-mediated isothermal amplification technology, serum agglutination tests, complement fixation tests, enzyme-linked immunosorbent assays, colloidal gold-labelled immunochromatography, and so on”. Please, include references here about these tests for brucellosis (even commercial examples).

3. “PCR, a diagnostic method for detecting human brucellosis, has the advantages of speed, safety, sensitivity, and specificity [7].” Are you sure speed is an advantage of PCR? Normally the main disadvantage of PCR is its time (plus costs and that it requires specific equipment and trained personnel).

4. “This technology should be considered to complement or confirm the traditional methods, i.e., serological methods or bacterial isolation and culture 50 methods [8].” If I’m not wrong PCR is a very accepted method, of high reliability and specificity, more than serological methods and cultures. Are you sure that PCR “should be considered to complement” and not as gold-standard method? (for example, for SARS-CoV-2 serological methods are used for fast tracking and confirmed by PCR). Please, review.

5. “Loop-mediated isothermal amplification (LAMP)”; please include a reference here about the technique (probably a good LAMP review paper).

6. “Mishra Adarsh designed two sets of LAMP primers for the omp2b gene of Brucella and standardized detection for all main Brucella strains.” Who is Mishra Adarash? Is he/she a famous scientist? Normally (except if this person is a celebrity and someone super popular), you would write “Mishra Adarsh et al.” or the name of the research group/laboratory.

7. The term “colloidal gold” is old fashioned. Please, check if you can use the term “gold nanoparticles” (if the dimension is under 100 nm). A typical abbreviation for gold nanoparticles would be AuNPs, which you can use.

8. You mention the advantages of RVFT but not the drawbacks. In the market, lateral flow tests stand over vertical flow tests (actually the number of publications about vertical flow is in decrease). Therefore, there should be some disadvantages in the technology. Could you identify? Why to develop a vertical and not a lateral flow?

9. “classic trisodium citrate is mostly used to prepare colloidal gold, which is marked and made into a detection card box”. I do not understand what do you mean with “marked”.

Preparation of gold nanoparticles

10. Please, indicate in this section the dimension of the nanomaterial.

11. “The prepared gold nanoparticle solution with a concentration of 10-5 was filtered…”. The units of concentration are missing (nanoparticles/mL? [Au]/L? moles of nanoparticles/dm^3?

12. “pH was adjusted to approximately 6.5 by using 0.2 mol/L K2CO3”. The K2CO3 salt cannot modify the pH, are you sure you used this salt? Or maybe some carbonate/hydrogencarbonate buffer?

13. Figure 2. Why the control is Sheep IgG? It is not explained in the text.

14. “First, sample dropping steps: the sample is serum sample or tissue fluid, etc., and the absorbed sample is directly dropped into the visible window of the filtration detection card. After the liquid is completely infiltrated, the color can be developed.” Why the color is visible before the addition of gold nanoparticles?

15. By reading the manuscript a second time, it is not clear what is the target being detected (LPS?), which is the bioreceptor for its recognition, etc. Text should be rewritten and probably Figure 2 replaced by a figure indicating what is placed in the nanomaterial, what on the nitrocellulose, etc.

16. “After AuCl- is reduced to elemental gold”. Are you sure you are reducing AuCl- and not AuCl4 1-?

17. “Figure 13. Experimental results of RVFT sensitivity”. I believe here you are testing your detection limit, not your sensitivity, please do not confuse both terms. In brief, limit of detection is the lowest concentration value you can detect. Sensitivity, in clinical or epidemiological field, is the % of true positives comparing your technique with a gold standard; or, in analytical field, the slope of the calibration curve of a quantitative method.

18. “First, RVFT shortens the detection time. The detection time of lateral flow immunochromatography is generally 15 to 20 minutes. However, the RVFT detection time is 2-3 minutes, which greatly shortens the detection time.” Actually I can find several publications about lateral flow assays which response is observed within 2-3 minutes, as RVFT, I do not think a RVFT assay is faster. Actually, in time calculation you are not considering the several buffers and solutions you are adding to the RVFT, while in LF is (in most of the cases) just a single solution. I guess that is why (going back to my question number 8) there are more lateral flow tests that reach the market and not the RVFT. Also, I would not agree with “It is difficult to use multiple lines in an lateral flow immunochroma- 532 tography sensor”, there are several publications with dots array in lateral flow (multiplexing) and several companies devoted to do it. The only point I agree is avoiding the hook effect, which is certain.

In general

19. The manuscript is too long (normally there is a limit of 6 figures and around 6000 words) and the structure some times is confusing. I would recommend moving figures and most of the optimization data to supporting information, focusing in the main text to the optimal conditions, results and analytical performance (also the analysis of analytical performance is missing; e.g. you could mesure the intensity of test dot at different concentrations and provide a calibration curve, detection limit, standard deviations, etc.)

20. I do not see many novelty in the manuscript. The gold nanoparticles synthesis is not new, the silver staining method is not new and the design of the RVFT is neither new. I cannot recommend the acceptance of the manuscript on its current form. I would recommend the rejection of this work but with the possibility to resubmit it in the future after rewriting the manuscript (not as a revision, but as a whole new paper).

Author Response

Response to Reviewer 2 Comments

Dear Editors and Reviewers:

Thank you for your letter and for the reviewers’ comments concerning our manuscript entitled

“Gold nanoparticles prepared with cyclodextrin applied to RVFT for the detection of brucellosis”

(Manuscript ID: biosensors-1782721). Those comments are all valuable and very helpful for revising

and improving our paper, as well as the important guiding significance to our researches. We have

studied comments carefully and have made correction which we hope meet with approval. In the

article, "Track Changes" was used to make corresponding modifications. The main corrections in

the paper and the responds to the reviewer’s comments are as following:

Abstract:

Point 1: I believe it should be concise. It’s not necessary to detail the optimization of gold

nanoparticles here; the sentence “The optimal conditions for preparing the gold nanoparticles were

as follows: the optimal reaction temperature was 200 °C, the optimal molar ratio of reactants was

1:6, the maximum appropriate amount of labelled staphylococcal protein A (SPA) was 600 µL, and

the maximum appropriate amount of blocking solution was 4 mL. The maximum concentration of

Tween-20 in the comprehensive buffer was 8%, and the A solution: B solution ratio in the silver

staining reagent was 1:3” can be removed.

Response 1: Thank you for your valuable comments. We have made corresponding changes based

on your comments. We have removed the optimization process detailing gold nanoparticles from

the abstract to keep the abstract concise.

Introduction

Point 2: “The diagnostic methods for brucellosis include PCR technology, nucleic acid probe

technology, loop-mediated isothermal amplification technology, serum agglutination tests,

complement fixation tests, enzyme-linked immunosorbent assays, colloidal gold-labelled

immunochromatography, and so on”. Please, include references here about these tests for

brucellosis (even commercial examples).

Response 2: Thank you very much for your advice. In fact, in the article, we have cited relevant

references at the end of each diagnosis of brucellosis.

Point 3: “PCR, a diagnostic method for detecting human brucellosis, has the advantages of speed,

safety, sensitivity, and specificity [7].” Are you sure speed is an advantage of PCR? Normally the

main disadvantage of PCR is its time (plus costs and that it requires specific equipment and trained

personnel).

Response 3: Thank you for your valuable comments. We also learned the method of PCR detection

of Brucellosis by referring to relevant references. In "PCR as a diagnostic tool for brucellosis", it is

written that "PCR has the potential to meet the need for better diagnostic tools. It is highly sensitive,

very specific, inexpensive, and easily adapted to high volume demands. The process is rapid ,

simple, and requires little manual labor.”[1] Betsy, J. B., PCR as a diagnostic tool for brucellosis. Veterinary Microbiology 2002, 90, (1).

Point 4: “This technology should be considered to complement or confirm the traditional methods,

i.e., serological methods or bacterial isolation and culture 50 methods [8].” If I’m not wrong PCR is

a very accepted method, of high reliability and specificity, more than serological methods and

cultures. Are you sure that PCR “should be considered to complement” and not as gold-standard

method? (for example, for SARS-CoV-2 serological methods are used for fast tracking and

confirmed by PCR). Please, review.

Response 4: Thank you for reading carefully. We reviewed a large amount of literature which

demonstrated that CFT is the standard method for detecting brucellosis. The specific verification

contents are as follows: ”The CFT is an international standard test used for detecting brucellosis.The

method used for the CFT was based on the one used by the OIE (World Organization for Animal

Health).”

Point 5: “Loop-mediated isothermal amplification (LAMP)”

; please include a reference here about

the technique (probably a good LAMP review paper).

Response 5: Thank you very much for your valuable advice. Based on your suggestion, we have

supplemented the appropriate references after reading a large number of relevant literature.

Point 6: “Mishra Adarsh designed two sets of LAMP primers for the omp2b gene of Brucella and

standardized detection for all main Brucella strains.” Who is Mishra Adarash? Is he/she a famous

scientist? Normally (except if this person is a celebrity and someone super popular), you would

write “Mishra Adarsh et al.” or the name of the research group/laboratory.

Response 6: Thank you for your valuable suggestions. We have made corresponding modifications

in the article.

Point 7: The term “colloidal gold” is old fashioned. Please, check if you can use the term “gold

nanoparticles” (if the dimension is under 100 nm). A typical abbreviation for gold nanoparticles

would be AuNPs, which you can use.

Response 7: Thank you for your valuable suggestions. We have made corresponding modifications

in the paper, changing colloidal gold to gold nanoparticles.

Point 8: You mention the advantages of RVFT but not the drawbacks. In the market, lateral flow

tests stand over vertical flow tests (actually the number of publications about vertical flow is in

decrease). Therefore, there should be some disadvantages in the technology. Could you identify?

Why to develop a vertical and not a lateral flow?

Response 8: Thank you for your valuable suggestions. The disadvantages of this technique are: 1.

Compared with the lateral flow immunochromatography technique, the method is relatively simple.

For example, lateral flow immunochromatography techniques include competition method,

sandwich method, indirect method and so on. Therefore, the substances that can be detected are

limited. 2. This technique requires additional synthesis buffer and chromogenic solution. This

technique uses vertical flow because the principle of percolation is used.Point 9: “classic trisodium citrate is mostly used to prepare colloidal gold, which is marked and

made into a detection card box”. I do not understand what do you mean with “marked”.

Response 9: We are very sorry for your confusion caused by the unclear meaning of our sentence.

In fact, what we want to express is to use gold nanoparticles prepared by trisodium citrate as the

marking material to mark target proteins for corresponding detection.

Preparation of gold nanoparticles

Point 10: Please, indicate in this section the dimension of the nanomaterial.

Response 10: Thank you very much for your valuable suggestions. We have made corresponding

notes in the result section.

Point 11: “The prepared gold nanoparticle solution with a concentration of 10-5 was filtered…”. The

units of concentration are missing (nanoparticles/mL? [Au]/L? moles of nanoparticles/dm^3?

Response 11: Thank you very much for your reminding. We have made corresponding

supplements.

Point 12: “pH was adjusted to approximately 6.5 by using 0.2 mol/L K2CO3”. The K2CO3 salt cannot

modify the pH, are you sure you used this salt? Or maybe some carbonate/hydrogencarbonate

buffer?

Response 12: Thank you very much for your question. We are pretty sure we are using K2CO3 in

this process. According to your question, we also checked relevant references. In the article

"Controlled copper in situ growth-amplified lateral flow sensors for sensitive, reliable, and

field-deployable infectious disease diagnostics", the author also uses K2CO3 to adjust PH—”In a

typical procedure, 1 mL citrate-coated AuNP solution was first adjusted to the appropriate pH with

0.01 M K2CO3.”

[1]Zhou Y.; Chen Y.; Liu Y.; Fang H.; Huang X.; Leng Y.; Liu Z.; Hou L.; Zhang W.; Lai W.; Xiong Y.,

Controlled copper in situ growth-amplified lateral flow sensors for sensitive, reliable, and

field-deployable infectious disease diagnostics. Biosensors and Bioelectronics, 2021, 171.

Point 13: Figure 2. Why the control is Sheep IgG? It is not explained in the text.

Response 13: Thank you very much for your valuable comments. The quality control point is used

to judge whether the test cartridge is valid. In this technique, recombinant Staphylococcus aureus

protein A is used as the labeled protein, because recombinant Staphylococcus aureus protein A has

high affinity and can bind to mammalian IgG Fc fragments. Therefore, the quality control points

can be coated with IgG.

Point 14: “First, sample dropping steps: the sample is serum sample or tissue fluid, etc., and the

absorbed sample is directly dropped into the visible window of the filtration detection card. After

the liquid is completely infiltrated, the color can be developed.” Why the color is visible before the

addition of gold nanoparticles?Response 14: We are very sorry that we misexpressed here. We have revised the text to "After the

liquid is completely soaked, the color development step can be performed."

Point 15: By reading the manuscript a second time, it is not clear what is the target being detected

(LPS?), which is the bioreceptor for its recognition, etc. Text should be rewritten and probably

Figure 2 replaced by a figure indicating what is placed in the nanomaterial, what on the

nitrocellulose, etc.

Response 15: Thank you very much for your valuable comments. LPS refers to highly purified

Brucella lipopolysaccharide. The position of the nitrocellulose membrane is clearly shown in Fig. 1,

and Fig. 2 is to show the spraying of T and C points in detail.

Point 16: “After AuCl- is reduced to elemental gold”. Are you sure you are reducing AuCl- and not

AuCl4 1-?

Response 16: Thank you very much for your valuable comments. We checked a lot of literature and

it was confirmed that “After AuCl- is reduced to elemental gold”.

Point 17: “Figure 13. Experimental results of RVFT sensitivity”. I believe here you are testing your

detection limit, not your sensitivity, please do not confuse both terms. In brief, limit of detection is

the lowest concentration value you can detect. Sensitivity, in clinical or epidemiological field, is

the % of true positives comparing your technique with a gold standard; or, in analytical field, the

slope of the calibration curve of a quantitative method.

Response 17: Thank you very much for your reminder. We have made corresponding changes in

the text according to your suggestions. Thank you again.

Point 18: “First, RVFT shortens the detection time. The detection time of lateral flow

immunochromatography is generally 15 to 20 minutes. However, the RVFT detection time is 2-3

minutes, which greatly shortens the detection time.” Actually I can find several publications about

lateral flow assays which response is observed within 2-3 minutes, as RVFT, I do not think a RVFT

assay is faster. Actually, in time calculation you are not considering the several buffers and

solutions you are adding to the RVFT, while in LF is (in most of the cases) just a single solution. I

guess that is why (going back to my question number 8) there are more lateral flow tests that reach

the market and not the RVFT. Also, I would not agree with “It is difficult to use multiple lines in an

lateral flow immunochroma- 532 tography sensor”, there are several publications with dots array in

lateral flow (multiplexing) and several companies devoted to do it. The only point I agree is

avoiding the hook effect, which is certain.

Response 18:

We are very sorry that we did not express clearly here, which caused ambiguity. For all the

operational detection times of lateral flow immunochromatography, we refer to the relevant

literature and commercial test strips that have been used in the detection of brucellosis using lateral

flow immunochromatography. We have made corresponding changes in the text.

Regarding the calculation of RVFT time, we are very sorry that we do not have a total time for the

entire detection process. After experimental verification, all operations and detection time of RVFTare generally 5-6 minutes. We have made corresponding changes in the text "All operations of

RVFT and detection time are generally 5-6 minutes".

Regarding "It is difficult to use multiple lines in an lateral flow immunochroma- 532 tography

sensor", we have made a mistake, and we have revised this part in the text. Thanks again for the

reminder.

Finally, thank you for your endorsement of avoiding the hook effect.

In general

Point 19: The manuscript is too long (normally there is a limit of 6 figures and around 6000 words)

and the structure some times is confusing. I would recommend moving figures and most of the

optimization data to supporting information, focusing in the main text to the optimal conditions,

results and analytical performance (also the analysis of analytical performance is missing; e.g. you

could mesure the intensity of test dot at different concentrations and provide a calibration curve,

detection limit, standard deviations, etc.)

Response 19: Thank you very much for your valuable suggestions. We have tried our best to revise

the article accordingly.

Point 20: I do not see many novelty in the manuscript. The gold nanoparticles synthesis is not new,

the silver staining method is not new and the design of the RVFT is neither new. I cannot

recommend the acceptance of the manuscript on its current form. I would recommend the rejection

of this work but with the possibility to resubmit it in the future after rewriting the manuscript (not

as a revision, but as a whole new paper).

Response 20:

So far, hybrid materials of macrocyclic supramolecules and gold nanoparticles have been widely

studied. On this type of gold nanoparticles, previous researchers have mainly focused on the

properties, functions and applications based on host-guest reactions. Considering the unique

structure of such macrocyclic supramolecules, their interaction with gold nanoparticles may

modulate the surface chemical energy leading to some new properties and corresponding potential

applications.

In this study, we synthesized cyclodextrin-functionalized gold nanoparticles. Cyclodextrin was

used as the only reducing agent and stabilizer. Without adding other reducing agents and

stabilizers, the green synthesis of cyclodextrin was carried out in one step. Modified gold

nanoparticles. We applied this gold nanoparticle to RVFT, hoping to provide a small force in the

detection technology of brucellosis.

Thank you for reviewing our article, which we acknowledge has many shortcomings. We have read

your detailed comments carefully and they are valuable. This is of great help to the revision and

improvement of this paper, and also has important guiding significance for our research. We did

our best to correct it and hope to be recognized. Thanks again for your diligent review and hope

you will give us another chance to improve. Thanks!

Thanks again!

We tried our best to improve the manuscript and made some changes in the manuscript. Thesechanges will not influence the content and framework of the paper. We used the "Track Changes"

function to make corresponding Changes, and the Changes are clearly visible.

We appreciate for Editors/Reviewers’ warm work earnestly, and hope that the correction will meet with

approval.

Once again, thank you very much for your comments and suggestions.

Sincerely yours

Feng Shi

Reviewer 3 Report

My suggestions are as follows,

1. Some of the sentences are complex and tedious to comprehend. I would recommend the authors to break them up and make it simpler.

2. How does author's work fare with the available technology? There is seldom comparison between what is proposed and how it is superior to the literature.

Author Response

Response to Reviewer 3 Comments

Dear Editors and Reviewers:

Thank you for your letter and for the reviewers’ comments concerning our manuscript entitled

“Gold nanoparticles prepared with cyclodextrin applied to RVFT for the detection of brucellosis”

(Manuscript ID: biosensors-1782721). Those comments are all valuable and very helpful for revising

and improving our paper, as well as the important guiding significance to our researches. We have

studied comments carefully and have made correction which we hope meet with approval. The

modified part of the article has been improved by using the "Track Change" tool. The main

corrections in the paper and the responds to the reviewer’s comments are as following:

Thank you very much for your recognition of our work, which gives us great encouragement. Of

course, we admit that there are many shortcomings in the manuscript. We have carefully read your

detailed comments, which are of great value. They are of great help to the revision and

improvement of our paper and also have important guiding significance for our research. We did

our best to make corrections and hope to be recognized.

Point 1: Some of the sentences are complex and tedious to comprehend. I would recommend the

authors to break them up and make it simpler.

Response 1: Thank you very much for your valuable suggestions. We have made corresponding

changes in the text.

Point 2: How does author's work fare with the available technology? There is seldom comparison

between what is proposed and how it is superior to the literature.

Response 2: Thank you very much for your valuable suggestions. We have added relevant content

to the text. The details are as follows: First, RVFT shortens the detection time. The detection time of

lateral flow immunochromatography is generally 10 to 15 minutes. However, the RVFT detection

time is 5-6 minutes, which greatly shortens the detection time. Second, RVFT effectively avoids the

hook effect [1], which is an interference associated with excess antigens that do not bind to the

conjugated antibody.

[1] Fernando, S.A.; Wilson, G.S., Studies of the 'hook' effect in the one-step sandwich immunoassay.

J Immunol Methods 1992, 151, (1-2), 47-66.

Thanks again!

We tried our best to improve the manuscript and made some changes in the manuscript. These

changes will not influence the content and framework of the paper. We used the "Track Changes"

function to make corresponding Changes, and the Changes are clearly visible.We appreciate for Editors/Reviewers’ warm work earnestly, and hope that the correction will meet with

approval.

Once again, thank you very much for your comments and suggestions.

Sincerely yours

Feng Shi

Round 2

Reviewer 2 Report

Dear authors, 

Thank you very much for considering my comments in the reviewed version of your work. I would like to comment few of your answers to my previous comments:

Old question 3: “PCR, a diagnostic method for detecting human brucellosis, has the advantages of speed, safety, sensitivity, and specificity [7].” Are you sure speed is an advantage of PCR? Normally the main disadvantage of PCR is its time (plus costs and that it requires specific equipment and trained personnel).

Authors response 3: Thank you for your valuable comments. We also learned the method of PCR detection of Brucellosis by referring to relevant references. In "PCR as a diagnostic tool for brucellosis", it is written that "PCR has the potential to meet the need for better diagnostic tools. It is highly sensitive, very specific, inexpensive, and easily adapted to high volume demands. The process is rapid, simple, and requires little manual labor.”[1] Betsy, J. B., PCR as a diagnostic tool for brucellosis. Veterinary Microbiology 2002, 90, (1)."

Comment on 3: Ok, the reference author says that his/her method is rapid, but we must be critic. In the manusctript is it explained how long takes the PCR? Please, check if it is explained because generally PCRs could last even hours, that won't be "rapid". I would consider fast something that can give you a response, including the sample collection, in less that 30 min.

Old question 8: You mention the advantages of RVFT but not the drawbacks. In the market, lateral flow tests stand over vertical flow tests (actually the number of publications about vertical flow is in decrease). Therefore, there should be some disadvantages in the technology. Could you identify? Why to develop a vertical and not a lateral flow?

Response 8: Thank you for your valuable suggestions. The disadvantages of this technique are: 1. Compared with the lateral flow immunochromatography technique, the method is relatively simple. For example, lateral flow immunochromatography techniques include competition method, sandwich method, indirect method and so on. Therefore, the substances that can be detected are limited. 2. This technique requires additional synthesis buffer and chromogenic solution. This technique uses vertical flow because the principle of percolation is used.

Comment on 8: Ok. Please, now I would like to see these drawbacks also explained in the manuscript. Your should be objective and always state the advantages and disavantages of your devices, so the readers can understand the limitations if they try to replicate or apply your work.

Old question 11: “The prepared gold nanoparticle solution with a concentration of 10-5 was filtered…”. The units of concentration are missing (nanoparticles/mL? [Au]/L? moles of nanoparticles/dm^3?

Response 11: Thank you very much for your reminding. We have made corresponding supplements.

Comment on 11: Ok, now that it is clear that the concentration is [Au]/mL, could you explain in the text how did you calculate/measure it? If it is calculated by approximating a 100% yield from the reaction, if it has been calculated from spectra (then it is not [Au] but [AuNPs], etc.)

Old question 13: Figure 2. Why the control is Sheep IgG? It is not explained in the text.

Response 13: Thank you very much for your valuable comments. The quality control point is used to judge whether the test cartridge is valid. In this technique, recombinant Staphylococcus aureus protein A is used as the labeled protein, because recombinant Staphylococcus aureus protein A has high affinity and can bind to mammalian IgG Fc fragments. Therefore, the quality control points can be coated with IgG.

Comment on 13: Please, explain it on section 3.5

Point 16: “After AuCl- is reduced to elemental gold”. Are you sure you are reducing AuCl- and not AuCl4 1-?

Response 16: Thank you very much for your valuable comments. We checked a lot of literature and it was confirmed that “After AuCl- is reduced to elemental gold”.

Comment on 16: HAuCl4 in water should dissociate to H+ and AuCl4, 1-. Then, AuCl4- will oxidize your reducing agent. Unless you can demonstrate an alternative chemical reaction in which AuCl4- is reduced to AuCl- before the final reaction, please write AuCl4-

Author Response

Response to Reviewer 2 Comments

Dear Editors and Reviewers:

Thank you for your letter and for the reviewers’ comments concerning our manuscript entitled “Gold nanoparticles prepared with cyclodextrin applied to RVFT for the detection of brucellosis” (Manuscript ID: biosensors-1782721). We know that there are a lot of shortcomings in this article and we need to continue to improve them. We thank you very much for your valuable comments. Those comments are all valuable and very helpful for revising and improving our paper, as well as the important guiding significance to our researches. We have studied comments carefully and have made correction which we hope meet with approval. The modified part of the article has been improved by using the "Track Change" tool. The main corrections in the paper and the responds to the reviewer’s comments are as following:

Old question 3: “PCR, a diagnostic method for detecting human brucellosis, has the advantages of speed, safety, sensitivity, and specificity [7].” Are you sure speed is an advantage of PCR? Normally the main disadvantage of PCR is its time (plus costs and that it requires specific equipment and trained personnel).

Authors response 3: Thank you for your valuable comments. We also learned the method of PCR detection of Brucellosis by referring to relevant references. In "PCR as a diagnostic tool for brucellosis", it is written that "PCR has the potential to meet the need for better diagnostic tools. It is highly sensitive, very specific, inexpensive, and easily adapted to high volume demands. The process is rapid, simple, and requires little manual labor.”[1] Betsy, J. B., PCR as a diagnostic tool for brucellosis. Veterinary Microbiology 2002, 90, (1)."

Comment on 3: Ok, the reference author says that his/her method is rapid, but we must be critic. In the manusctript is it explained how long takes the PCR? Please, check if it is explained because generally PCRs could last even hours, that won't be "rapid". I would consider fast something that can give you a response, including the sample collection, in less that 30 min.

Response 3: Thank you very much for your valuable suggestions. We have made corresponding modifications in the article according to your suggestions.

Old question 8: You mention the advantages of RVFT but not the drawbacks. In the market, lateral flow tests stand over vertical flow tests (actually the number of publications about vertical flow is in decrease). Therefore, there should be some disadvantages in the technology. Could you identify? Why to develop a vertical and not a lateral flow?

Response 8: Thank you for your valuable suggestions. The disadvantages of this technique are: 1. Compared with the lateral flow immunochromatography technique, the method is relatively simple. For example, lateral flow immunochromatography techniques include competition method, sandwich method, indirect method and so on. Therefore, the substances that can be detected are limited. 2. This technique requires additional synthesis buffer and chromogenic solution. This technique uses vertical flow because the principle of percolation is used.

Comment on 8: Ok. Please, now I would like to see these drawbacks also explained in the manuscript. Your should be objective and always state the advantages and disavantages of your devices, so the readers can understand the limitations if they try to replicate or apply your work.

Response 8: Thank you very much for your valuable suggestions. We have made corresponding supplements in the article.

Old question 11: “The prepared gold nanoparticle solution with a concentration of 10-5 was filtered…”. The units of concentration are missing (nanoparticles/mL? [Au]/L? moles of nanoparticles/dm^3?

Response 11: Thank you very much for your reminding. We have made corresponding supplements.

Comment on 11: Ok, now that it is clear that the concentration is [Au]/mL, could you explain in the text how did you calculate/measure it? If it is calculated by approximating a 100% yield from the reaction, if it has been calculated from spectra (then it is not [Au] but [AuNPs], etc.)

Response 11: Thank you very much for your valuable comments. We calculated this with a reaction yield close to 100%. The corresponding calculation is carried out by the addition amount of chloroauric acid. However, after a lot of literature review and multiple inquiries, the concentration unit cannot be changed to AuNPs/mL. In order not to cause ambiguity, we have made corresponding changes in the text. In Section 3.1, the preparation scheme of the gold nanoparticle solution and the molar ratio of HAuCl4 and β-cyclodextrin are written clearly.

Old question 13: Figure 2. Why the control is Sheep IgG? It is not explained in the text.

Response 13: Thank you very much for your valuable comments. The quality control point is used to judge whether the test cartridge is valid. In this technique, recombinant Staphylococcus aureus protein A is used as the labeled protein, because recombinant Staphylococcus aureus protein A has high affinity and can bind to mammalian IgG Fc fragments. Therefore, the quality control points can be coated with IgG.

Comment on 13: Please, explain it on section 3.5

Response 13: Thank you very much for your reminding. We have made corresponding additions at appropriate places in the text. 

Point 16: “After AuCl- is reduced to elemental gold”. Are you sure you are reducing AuCl- and not AuCl4 1-?

Response 16: Thank you very much for your valuable comments. We checked a lot of literature and it was confirmed that “After AuCl- is reduced to elemental gold”.

Comment on 16: HAuCl4 in water should dissociate to H+ and AuCl4, 1-. Then, AuCl4- will oxidize your reducing agent. Unless you can demonstrate an alternative chemical reaction in which AuCl4- is reduced to AuCl- before the final reaction, please write AuCl4-

Response 16: We are very sorry that we cannot prove this chemical reaction. According to your suggestion, we have made corresponding modifications in the article.

Finally, thank you again for your many revisions to our article. It is that we have learned a lot of new knowledge from it, and also improved the article, which also has important guiding significance for our research. We have tried our best to correct it, and hope to get your valuable recognition and give us another chance to improve. Thanks!

Thanks again!

We tried our best to improve the manuscript and made some changes in the manuscript.  These changes will not influence the content and framework of the paper. We used the "Track Changes" function to make corresponding Changes, and the Changes are clearly visible.
We appreciate for Editors/Reviewers’ warm work earnestly, and hope that the correction will meet with approval.
Once again, thank you very much for your comments and suggestions.

Sincerely yours

Feng Shi

This manuscript is a resubmission of an earlier submission. The following is a list of the peer review reports and author responses from that submission.

Round 1

Reviewer 1 Report

Manuscript biosensors-1736615 by Fang et al. describes a membrane-based, rapid vertical flow immunoassay labeled with colloidal nanoparticles for Brucella detection. After reading the manuscript carefully, I must give a negative comment. The manuscript describes a routine optimization of the Au-labeled immunoassay, with no clear elements of novelty. The selection of the results presented in the manuscript (very basic research), the manner of their description and the conclusions drawn are often questionable. The manuscript is quite chaotic - Introduction contains basic descriptions of various analytical methods, while in the Results section, elements typical for the introduction section are inserted in random places. Despite the average volume of the manuscript (12 p.), the manuscript is sparse in detail and filled with basic information and optimizations. The results  are rather poorly described from the analytical point of view - there is no statistical information, detailed characterization of analytical performance as well as their comparison with the already described methods. In my opinion, this manuscript should not be considered for publication in the Biosensors journal.

Please see my detailed comments below:

1) Please explain abbreviations on first use (RVFT, SPA, NC etc.) - please avoid unexplained abbreviation in the title.

2) The description of various methods of brucellosis detection in Introduction is very extensive and basic - it would be better to focus on comparing analytical parameters of each of the methods (taking into account the journal profile, it would be highly advisable).

3) Chapter 2.3: The synthesis description is not very detailed and contains confusing mental abbreviations - eg how to heat a 1% HAuCl4 solution (aqueous, as I expect) to 200oC (according to provided description) while keeping the form of a colloidal solution?

4) Line 61 - what is "the complex solution" added to the supernatant?

5)Please justify the use of cyclodextrin to stabilize NPs earlier in the text. Nanoparticles stabilized with typical ligands (e.g. citrate) seem to be just as good or even better as a substrate for passive adsorption of SPA...

6) What is the mechanism of catalytic activity of cyclodextrin towards silver staining? Was the effect of dextrin alone or only in combination with AuNPs tested?

7) Is the synthesis of gold nanoparticles reproducible? Were the presented optimization studies based on a single experiment or a series? Were the results compared between synthetic batches?

8) I believe that Figures 4, 5 and 6 convey very similar information and could be combined into a single one (Figure 6 duplicates the spectrum presented in Figure 4).

9) The conclusions of chapters 3.3 and 3.4 are not supported by any experimental results.

10) Chapter 3.9: what was the incubation time during silver staining (this information is missing)

11) Paragraph describing cyclodextrins (at the end of the Results section) seems to be completely in the wrong place - please consider moving it, eg. to the introduction section.

Author Response

Response to Reviewer 1 Comments

Dear Editors and Reviewers:

Thank you for your letter and for the reviewers’ comments concerning our manuscript entitled “Gold nanoparticles prepared with cyclodextrin applied to RVFT for the detection of brucellosis” (Manuscript ID: biosensors-1736615). We know that there are a lot of shortcomings in this article and we need to continue to improve them. We thank you very much for your valuable comments. Those comments are all valuable and very helpful for revising and improving our paper, as well as the important guiding significance to our researches. We have studied comments carefully and have made correction which we hope meet with approval. The modified part of the article has been improved by using the "Track Change" tool. The main corrections in the paper and the responds to the reviewer’s comments are as following:

General comment:

Manuscript biosensors-1736615 by Fang et al. describes a membrane-based, rapid vertical flow immunoassay labeled with colloidal nanoparticles for Brucella detection. After reading the manuscript carefully, I must give a negative comment. The manuscript describes a routine optimization of the Au-labeled immunoassay, with no clear elements of novelty. The selection of the results presented in the manuscript (very basic research), the manner of their description and the conclusions drawn are often questionable. The manuscript is quite chaotic - Introduction contains basic descriptions of various analytical methods, while in the Results section, elements typical for the introduction section are inserted in random places. Despite the average volume of the manuscript (12 p.), the manuscript is sparse in detail and filled with basic information and optimizations. The results  are rather poorly described from the analytical point of view - there is no statistical information, detailed characterization of analytical performance as well as their comparison with the already described methods. In my opinion, this manuscript should not be considered for publication in the Biosensors journal.

Response: Thank you for reviewing our article, which we acknowledge has many shortcomings. We have read your detailed comments carefully and they are valuable. This is of great help to the revision and improvement of this paper, and also has important guiding significance for our research. We did our best to correct it and hope to be recognized. Thanks again for your diligent review and hope you will give us another chance to improve. Thanks!

Point 1: Please explain abbreviations on first use (RVFT, SPA, NC etc.) - please avoid unexplained abbreviation in the title.

Response 1: We are very sorry that the first use of abbreviations (RVFT, SPA, NC, etc.) did not give the full terminology. We have made corresponding additions in the title and text. Thank you very much for reminding.

Point 2: The description of various methods of brucellosis detection in Introduction is very extensive and basic - it would be better to focus on comparing analytical parameters of each of the methods (taking into account the journal profile, it would be highly advisable).

Response 2: Thank you very much for your valuable advice. We have made corresponding improvements in the paper.

The additions in the article are as follows:

The sensitivity of CFT has been reported to be lower in sheep in field conditions (88.6%) than those of RBT (92.1%) and iELISA (100%) . 

  1. Blasco, J.M., Garin-Bastuji, B., Mar ın, C.M., Gerbier, G., Fanlo, J., Jim enez de Bag u es, M.P., Cau, C., Efficacy of different rose bengal and complement fixation antigens for the diagnosis of Brucella melitensis in sheep and goats. Vet. Rec. 1994a, 134, 415 420.
  2. Blasco, J.M., Mar ın, C., Jim enez de Bag u es, M.P., Barber an, M., Hern andez, A., Molina, L., Velasco, J., D ıaz, R., Moriy on, I., Evaluation of allergic and serological tests for diagnosing Brucella melitensis infection in sheep. J. Clin. Microbiol. 1994b, 32, 1835 1840. 

Point 3: Chapter 2.3: The synthesis description is not very detailed and contains confusing mental abbreviations - eg how to heat a 1% HAuCl4 solution (aqueous, as I expect) to 200oC (according to provided description) while keeping the form of a colloidal solution?

Response 3: We are very sorry that this section is not described in detail. We have revised this section, hoping to make Chapter 2.3 clear.

The revisions in the article are as follows:

At 200℃, 1% chlorauric acid solution was added to 50 mL ultra-pure water at 600 RPM and stirred for 5 minutes. Quickly add β -cyclodextrin and 120 μL NaOH. When the solution turns wine red, stop heating and stir for another 10 minutes. Colour change: colourless-light pink-purple-burgundy.

Point 4: Line 61 - what is "the complex solution" added to the supernatant?

Response 4: We are very sorry that this section is not described in detail. We have explained "the complex solution" in the text.

The corresponding supplement in the article is as follows:

(The complex solution consisted of 1mL of 5% bovine serum albumin, 500μL of 5% sucrose, 500μL of 5% trehalose, 1mL of 10% polyvinylpyrrolidone, and 7mL of Tris HCl, pH 8)

Point 5: Please justify the use of cyclodextrin to stabilize NPs earlier in the text. Nanoparticles stabilized with typical ligands (e.g. citrate) seem to be just as good or even better as a substrate for passive adsorption of SPA...

Response 5: Thank you very much for your valuable advice. Earlier in the text, we have emphasized the importance of cyclodextrin and made corresponding improvements.

The supplementary content is as follows: 

Cyclodextrins have remarkable capability to establish supramolecular host guest interactions because of their toroidal shape and non-polar inside [1,2]. The less hydrophilic internal cavity and the hydrophilic exterior ofβ-CD enable it to make supramolecular interactions such as electrostatic, hydrogen bonding, van der Waals forces and hydrophobic interactions with organic, inorganic and bio molecules [3]. Cyclodextrin molecules contribute distinguished advantages due to their novel architectural features to form inclusion complexes with several kinds of molecules like ions, protein, and oligonucleotides [4].

Based on its important physical and chemical properties, cyclodextrin was used as a reducing agent to prepare gold nanoparticles. Different from previous reports, the cyclodextrin-modified gold nanoparticles designed by us are synthesized in one step without the addition of other reducing agents and modifiers. This method not only provides a new direction for chemical synthesis, but also provides a diversified direction for the application of gold nanoparticles.

  1. 1. Nadia, M.; Sophie, F.; Éva, F.; Eric, L.; Giangiacomo, T.; Marc, F.; Grégorio, C., 130 years of cyclodextrin discovery for health, food, agriculture, and the industry: a review. Environ Chem Lett 2021, (prepublish).
  2. 2. Max, P.; Iñigo, X.G.; José, R.I., History of cyclodextrin-based polymers in food and pharmacy: a review. Environ Chem Lett 2021, (prepublish).
  3. 3. , A.C., ChemInform Abstract: The Stability of Cyclodextrin Complexes in Solution. ChemInform 1997, 28, (44).
  4. 4. Melanie, A.L.; Susanna, W., Innovations in Oligonucleotide Drug Delivery. J Pharm Sci-Us 2003, 92, (8).

Point 6: What is the mechanism of catalytic activity of cyclodextrin towards silver staining? Was the effect of dextrin alone or only in combination with AuNPs tested?

Response 6: Thank you very much for your valuable suggestions. The catalytic activity mechanism of cyclodextrin on silver dyeing is as follows. Cyclodextrin is combined with AuNPs.

The catalytic activity mechanism of cyclodextrin on silver dyeing is as follows:

The principle of the technique is to label SPA with gold nanoparticles prepared by cyclodextrin and then introduce silver ions into the system. The small size effect of colloidal gold particles makes it have a strong catalytic reduction ability, which can reduce silver ions in the surrounding system to silver particles. These silver particles can further catalyze the reduction of silver ions in the system. This cascade-like catalytic action makes the silver ions gather more and more, tightly wrap the gold nanoparticles, and finally accumulate into agglomerated silver shells, forming black particles visible to the naked eye.

Point 7: Is the synthesis of gold nanoparticles reproducible? Were the presented optimization studies based on a single experiment or a series? Were the results compared between synthetic batches?

Response 7: Thank you very much for your valuable comments. We consider the synthesis of gold nanoparticles to be non-renewable. Our proposed optimization study is based on a series of experiments. We compared results on synthetic batches. Details can be found in the text.

Point 8: I believe that Figures 4, 5 and 6 convey very similar information and could be combined into a single one (Figure 6 duplicates the spectrum presented in Figure 4).

Response 8: Thank you very much for your valuable comments. What you think makes sense, but we added Figure 6 to make the characterization results of the best quality gold nanoparticles more clearly presented to the reader. We have modified Figures 4 and 5 accordingly.

Point 9: The conclusions of chapters 3.3 and 3.4 are not supported by any experimental results.

Response 9: Thank you very much for your valuable comments. Regarding Chapter 3.3, we put it here to emphasize that NC films can greatly affect the experimental results. Regarding Chapter 3.4, we put it in this position to emphasize the importance of antigen selection. We have made corresponding changes in the text and moved these two chapters to the Materials and Methods section.

Point 10: Chapter 3.9: what was the incubation time during silver staining (this information is missing)

Response 10: We apologize for the loss of incubation time during silver staining. For this information, we have made corresponding improvements in the article.

The revisions in the article are as follows:

Plot of incubation time results for silver staining

Reaction time of the process of silver staining was optimized here, as Figure 13 clarified. The incubation time during silver dyeing was set as gradient, divided into 1 minute, 2 minutes, 3 minutes, 4 minutes and 6 minutes. Experimental results show that when the incubation time is less than 3 minutes, the time was too short to make enough silver ions transformed into silver metal, leading to inconspicuous color change of the strip. But when the reaction time increased to 6 min, metallic silver produced by the process was enormous to be detected outside T point and C point on the membrane. The suitable incubation time was 3 minutes. AuNPs could be fully covered by silver ions and resulted in distinct black color visualized on T point and C point without diffusion.

Point 11: Paragraph describing cyclodextrins (at the end of the Results section) seems to be completely in the wrong place - please consider moving it, eg. to the introduction section.

Response 11: Thank you very much for your valuable comments. We have moved this paragraph to the Introduction section.

Thanks again!

We tried our best to improve the manuscript and made some changes in the manuscript.  These changes will not influence the content and framework of the paper. We used the "Track Changes" function to make corresponding Changes, and the Changes are clearly visible.
We appreciate for Editors/Reviewers’ warm work earnestly, and hope that the correction will meet with approval.
Once again, thank you very much for your comments and suggestions.

Sincerely yours

Feng Shi

Reviewer 2 Report

The authors reported a rapid vertical flow technology (RVFT) device for the detection of brucellosis made of β-Cyclodextrin-functionalized gold nanoparticles and amplified by the silver staining technique. This approach showed good detection sensitivity, low coast, simplified operating procedures, and no sample pretreatment was required. However, there are certain concerns about novelty and experiments. Therefore, I will reconsider accepting this work once the following questions are well addressed.

  1. Cyclodextrin plays a minor part in this project because the authors only need gold NPs for the RVFT. Gold NPs synthesized with traditional trisodium citrate can also do this function. Please stress the importance of cyclodextrin.
  2. The authors mentioned SPA without giving its entire term in the abstract and introduction, and I had no idea what it meant until I saw SPA in the Experimental Reagents section. Furthermore, the authors provide the best conditions for making gold NPs, however they do not provide precise results on sensitivity or detection limit decrease in the abstract part.
  3. The author goes into great detail about brucellosis detection methods such as PCR, LAMP, SAT, CFT, and so on in the introduction section. However, there is less information regarding RVFT in earlier research.
  4. The authors mentioned that the gold NPs were stable after 6 months (line 260) and that the Tindal effect appeared with these NPs (line 263), however there were no figures to back up their claims.
  5. Please double-check the manuscript for typos. In line 320, for example, "Optimization of marker SAP" should be "SPA".
  6. The authors' RVFT detection approach is unclear. More details are needed, such as the gold concentration employed, the incubation time of gold nanoparticle-labeled SPA, Tween, or silver staining solution, and so on.
  7. The authors claimed that their method had a quick detection speed, however they did not specify how much time it took.

Author Response

Response to Reviewer 2 Comments

Dear Editors and Reviewers:

Thank you for your letter and for the reviewers’ comments concerning our manuscript entitled “Gold nanoparticles prepared with cyclodextrin applied to RVFT for the detection of brucellosis” (Manuscript ID: biosensors-1736615). Those comments are all valuable and very helpful for revising and improving our paper, as well as the important guiding significance to our researches. We have studied comments carefully and have made correction which we hope meet with approval. In the article, "Track Changes" was used to make corresponding modifications. The main corrections in the paper and the responds to the reviewer’s comments are as following:

General comment:

The authors reported a rapid vertical flow technology (RVFT) device for the detection of brucellosis made of β-Cyclodextrin-functionalized gold nanoparticles and amplified by the silver staining technique. This approach showed good detection sensitivity, low coast, simplified operating procedures, and no sample pretreatment was required. However, there are certain concerns about novelty and experiments. Therefore, I will reconsider accepting this work once the following questions are well addressed.

Response: Thank you for your interest in our work. We acknowledge that there are many inadequacies in the manuscript and we will try our best to make them up. At the same time, thank you for your valuable suggestions.

We have carefully read your detailed comments, which are of great value. They are of great help to the revision and improvement of our paper, and also have important guiding significance for our research. We did our best to make corrections and hope to be recognized.

Point 1: Cyclodextrin plays a minor part in this project because the authors only need gold NPs for the RVFT. Gold NPs synthesized with traditional trisodium citrate can also do this function. Please stress the importance of cyclodextrin. 

Response 1: Thank you for your valuable suggestions. In the introduction, we have emphasized the importance of cyclodextrin and made corresponding improvements.

The supplementary content is as follows: 

Cyclodextrins have remarkable capability to establish supramolecular host guest interactions because of their toroidal shape and non-polar inside [1,2]. The less hydrophilic internal cavity and the hydrophilic exterior of β-CD enable it to make supramolecular interactions such as electrostatic, hydrogen bonding, van der Waals forces and hydrophobic interactions with organic, inorganic and bio molecules [3]. Cyclodextrin molecules contribute distinguished advantages due to their novel architectural features to form inclusion complexes with several kinds of molecules like ions, protein, and oligonucleotides [4].

Based on its important physical and chemical properties, cyclodextrin was used as a reducing agent to prepare gold nanoparticles. Different from previous reports, the cyclodextrin-modified gold nanoparticles designed by us are synthesized in one step without the addition of other reducing agents and modifiers. This method not only provides a new direction for chemical synthesis, but also provides a diversified direction for the application of gold nanoparticles.

  1. 1. Nadia, M.; Sophie, F.; Éva, F.; Eric, L.; Giangiacomo, T.; Marc, F.; Grégorio, C., 130 years of cyclodextrin discovery for health, food, agriculture, and the industry: a review. Environ Chem Lett 2021, (prepublish).
  2. 2. Max, P.; Iñigo, X.G.; José, R.I., History of cyclodextrin-based polymers in food and pharmacy: a review. Environ Chem Lett 2021, (prepublish).
  3. 3. , A.C., ChemInform Abstract: The Stability of Cyclodextrin Complexes in Solution. ChemInform 1997, 28, (44).
  4. 4. Melanie, A.L.; Susanna, W., Innovations in Oligonucleotide Drug Delivery. J Pharm Sci-Us 2003, 92, (8).

Point 2: The authors mentioned SPA without giving its entire term in the abstract and introduction, and I had no idea what it meant until I saw SPA in the Experimental Reagents section. Furthermore, the authors provide the best conditions for making gold NPs, however they do not provide precise results on sensitivity or detection limit decrease in the abstract part.

Response 2: We are very sorry that the complete terms of SPA (Staphylococcal protein A) were not given in the abstract and introduction. We have made corresponding supplements in the article. Thank you very much for your reminding. We have added the precise results of the decrease in sensitivity and detection limit in the abstract.

The changes in the abstract are as follows:

Standard Brucella-Positive Serum (containing Brucella antibody at 4000 IU/mL) could be detected in this system even for a dilution factor of 1×10-3. The detection limit was 4 IU/mL.

Point 3: The author goes into great detail about brucellosis detection methods such as PCR, LAMP, SAT, CFT, and so on in the introduction section. However, there is less information regarding RVFT in earlier research.

Response 3: Thank you for your valuable suggestions. In the introduction, we have added relevant information about RVFT in earlier studies.

The following is added in the introduction: 

RVFT was originally established by Spielberg et al. in 1989 for the detection of antibodies to human immunodeficiency virus[1]. RVFT based on nitrocellulose membrane reaction carrier, the use of microporous membrane filter, the immune response and washing in the special percolation device by means of liquid permeability filtration membrane is complete, the process has the affinity chromatography enrichment effect, can reach the purpose of rapid detection, with colloidal gold as a biomarker for color make positive results on the carrier film appears as red spots, The naked eye can make qualitative judgment on the detection results, and the detection process is easy to operate without any auxiliary instruments and equipment, which is widely used in food detection, clinical disease detection and animal disease detection, etc. [2-4].

  1. Spielberg, F.; Ryder, R.; Harris, J.; Heyward, W.; Kabeya, C.; Kifuani, N.K.; Bender, T.; Quinn, T., FIELD TESTING AND COMPARATIVE EVALUATION OF RAPID, VISUALLY READ SCREENING ASSAYS FOR ANTIBODY TO HUMAN IMMUNODEFICIENCY VIRUS. The Lancet 1989, 333, (8638).
  2. Debjani, S.; Dipika, R.; Tarun, K.D., Immunofiltration assay for aflatoxin B 1 based on the separation of pre-immune complexes. J Immunol Methods 2013, 392, (1-2).
  3. Zhang, P.; Bao, Y.; Draz, M.S.; Lu, H.; Liu, C.; Han, H., Rapid and quantitative detection of C-reactive protein based on quantum dots and immunofiltration assay. Int J Nanomed 2015, 10, (default).
  4. , A.B.; T., Y.R.; N., A.Y.; S., S.; I., Y.G., Simultaneous Determination of Several Mycotoxins by Rapid Immunofiltration Assay. J Anal Chem.2014, 69, (6).

Point 4: The authors mentioned that the gold NPs were stable after 6 months (line 260) and that the Tindal effect appeared with these NPs (line 263), however there were no figures to back up their claims.

Response 4: We are very sorry for the lack of supporting information of these two parts. We have made relevant supplements in the article.

Here's our supplementary chart:

Table 1. Stability observation results of gold nanoparticles

Gold nanoparticle solution

Time (month)

1

2

3

4

5

6

7

Result

-

-

-

-

-

-

+-

1Note: - means no precipitation, +- weak precipitation

Figure 5. (a) Image of the gold nanoparticle solution; (b)Tindal effect diagram.

Point 5: Please double-check the manuscript for typos. In line 320, for example, "Optimization of marker SAP" should be "SPA".

Response 5: We are very sorry for the mistake. We have gone through the whole article carefully and corrected the spelling mistakes accordingly.

Point 6: The authors' RVFT detection approach is unclear. More details are needed, such as the gold concentration employed, the incubation time of gold nanoparticle-labeled SPA, Tween, or silver staining solution, and so on.

Response 6: Thank you very much for your valuable comments. For the RVFT detection method, we have made corresponding improvements in the text. We have supplemented accordingly in the details of the gold concentration employed, the incubation time of gold nanoparticle-labeled SPA,Tween, or silver staining solution, and so on.

The optimal silver staining incubation time was determined as follows:

Reaction time of the process of silver staining was optimized here, as Figure 13 clarified. The incubation time during silver dyeing was set as gradient, divided into 1 minute, 2 minutes, 3 minutes, 4 minutes and 6 minutes. Experimental results show that when the incubation time is less than 3 minutes, the time was too short to make enough silver ions transformed into silver metal, leading to inconspicuous color change of the strip. But when the reaction time increased to 6 min, metallic silver produced by the process was enormous to be detected outside T point and C point on the membrane. The suitable incubation time was 3 minutes. AuNPs could be fully covered by silver ions and resulted in distinct black color visualized on T point and C point without diffusion.

Point 7: The authors claimed that their method had a quick detection speed, however they did not specify how much time it took.

Response 7: Thank you for your valuable suggestions.We have made corresponding supplements in the article.

The following was added to the summary:

RVFT is simple to operate, short reaction time, 2-3 minutes visible to the naked eye, without any equipment. 

Thanks again!

We tried our best to improve the manuscript and made some changes in the manuscript.  These changes will not influence the content and framework of the paper. We used the "Track Changes" function to make corresponding Changes, and the Changes are clearly visible.

We appreciate for Editors/Reviewers’ warm work earnestly, and hope that the correction will meet with approval.
Once again, thank you very much for your comments and suggestions.

Sincerely yours

Feng Shi

Reviewer 3 Report

In this work, the paper exhibits development of a convenient, fast, effective, and inexpensive detection method for brucellosis. Gold nanoparticles labeled with β-cyclodextrin as a reducing agent were synthesized and optimized, applied to rapid vertical flow technology (RVFT), and used to establish a kit for the tracing of brucellosis. The optimal conditions for preparing the gold nanoparticles were as follows: the optimal reaction temperature was 200 °C, the optimal molar ratio of reactants was 1:6, the maximum appropriate amount of labeled SPA was 600 μL, and the maximum appropriate amount of blocking solution was 4 mL. The maximum concentration of Tween-20 in the comprehensive buffer was 8%, and the A solution: B solution ratio in the silver staining reagent was 1:3. In this study, gold nanoparticles prepared from β-cyclodextrin were applied to RVFT for the first time, and on this basis, silver staining amplification technology was introduced, which further improved the sensitivity and reduced the detection limit of this method. Therefore, I recommend this manuscript for publication in molecules after major revision.

Introduction:

  • Insert a new paragraph to explain the advantages of the rapid vertical flow technology sensing technique with respect to other utilizing techniques?
  • Clarify the benefits of (rapid vertical flow technology) make it the desired choice than others?
  • provide short notes about the applied mechanism and compare this mechanism with other utilized mechanisms?

Results and discussion

  • The advantages of this method in comparison with other methods should be highlighted, including analytical characteristics, reproducibility, specificity, and stability?
  • No data about reversibility are given?
  • With respect to the preparation method and the effect of temperature: 1-There is a significant relationship between the absorbance spectra and the size of the NPs (provide an explanation and insert suitable citations)                                                                                                  2-It will be better to have a photo of the prepared solutions (to clarify the different colors)                                                                                 3- The mechanism should be explained based on temperature
  • Provide more details about the function and choice of the Tween-20 surfactant.
  • The validation of this technique should be introduced by comparison with a previously validated method.

Author Response

Response to Reviewer 3 Comments

Dear Editors and Reviewers:

Thank you for your letter and for the reviewers’ comments concerning our manuscript entitled “Gold nanoparticles prepared with cyclodextrin applied to RVFT for the detection of brucellosis” (Manuscript ID: biosensors-1736615). Those comments are all valuable and very helpful for revising and improving our paper, as well as the important guiding significance to our researches. We have studied comments carefully and have made correction which we hope meet with approval. The modified part of the article has been improved by using the "Track Change" tool. The main corrections in the paper and the responds to the reviewer’s comments are as following:

General comment:

In this work, the paper exhibits development of a convenient, fast, effective, and inexpensive detection method for brucellosis. Gold nanoparticles labeled with β-cyclodextrin as a reducing agent were synthesized and optimized, applied to rapid vertical flow technology (RVFT), and used to establish a kit for the tracing of brucellosis. The optimal conditions for preparing the gold nanoparticles were as follows: the optimal reaction temperature was 200 °C, the optimal molar ratio of reactants was 1:6, the maximum appropriate amount of labeled SPA was 600 μL, and the maximum appropriate amount of blocking solution was 4 mL. The maximum concentration of Tween-20 in the comprehensive buffer was 8%, and the A solution: B solution ratio in the silver staining reagent was 1:3. In this study, gold nanoparticles prepared from β-cyclodextrin were applied to RVFT for the first time, and on this basis, silver staining amplification technology was introduced, which further improved the sensitivity and reduced the detection limit of this method. Therefore, I recommend this manuscript for publication in molecules after major revision.

Response: Thank you very much for your recognition of our work, which gives us great encouragement. Of course, we admit that there are many shortcomings in the manuscript. We have carefully read your detailed comments, which are of great value. They are of great help to the revision and improvement of our paper and also have important guiding significance for our research. We did our best to make corrections and hope to be recognized.

Introduction

Point 1: Insert a new paragraph to explain the advantages of the rapid vertical flow technology sensing technique with respect to other utilizing techniques?

Response 1: Thank you very much for your valuable advice. In the introduction, we have inserted a new paragraph to explain the advantages of fast vertical flow sensing technology over other utilization technologies.

The main advantages of RVFT are as follows:

  • RVFT is easy to use and simple to operate, and users do not need special training.

(2) RVFT is safe, non-toxic, and has no side effects. Because colloidal gold itself has color, it does not need to add color developer and stop solution, and avoids the use of radioisotopes and toxic substrates/organisms.

(3) RVFT is economical and environmentally friendly: the sample basically does not need to be pretreated, and can be tissue fluid, serum and urine, eliminating the need for sample pretreatment in traditional analysis methods.

(4) RVFT detection time is short: RVFT can generally get the detection result in 2~3min, which is unattainable by other detection methods at present.

(5) The detection cartridge is small in size and easy to carry: RVFT's detection cartridge is suitable for field operations or on-site temporary detection, and is not limited by the experimental site and conditions.

(6) RVFT production cost and detection cost are both low: the amount of reagents and samples is small, and the sample volume can be as low as 10 μL, which reduces the cost of detection; at the same time, the materials for preparing the detection cartridge are cheap, and large-scale equipment is not required, and the production cost is very low.

(7) The experimental results can be stored for a long time: the prepared strip can be stored for one year or even longer in the refrigerator at 4℃.

(8) There are many types of test specimens: RVFT can be used to test tissue fluid, urine or feces, etc., so it is suitable for various tests.

(9) RVFT has a wide range of applications and can be used for detection in many clinical and non-clinical fields.

Point 2: Clarify the benefits of (rapid vertical flow technology) make it the desired choice than others?

Response 2: Thank you very much for your valuable comments. In the introduction section, we have made a corresponding statement that RVFT is a more ideal choice than other techniques.

Point 3: provide short notes about the applied mechanism and compare this mechanism with other utilized mechanisms?

Response 3: Thank you very much for your valuable comments. We have added a brief description of the applied mechanism to the text and compared this mechanism to other mechanisms used.

The supplement in the article is as follows:

The following is added in the introduction:

RVFT was originally established by Spielberg et al. in 1989 for the detection of antibodies to human immunodeficiency virus[1]. RVFT uses NC film as the reaction carrier, and uses the filterability of microporous membrane to complete the immune reaction and washing on a special diafiltration device through liquid diafiltration through the membrane. This process has an affinity chromatography. Concentration can achieve the purpose of rapid detection. Using colloidal gold as a chromogenic marker makes the positive results appear as red spots on the carrier film, and the detection results can be qualitatively judged with the naked eye. The detection process is easy to operate and does not require any auxiliary equipment. , is widely used in food testing, clinical disease testing and animal disease testing and other fields [2-4].

  1. Spielberg, F.; Ryder, R.; Harris, J.; Heyward, W.; Kabeya, C.; Kifuani, N.K.; Bender, T.; Quinn, T., FIELD TESTING AND COMPARATIVE EVALUATION OF RAPID, VISUALLY READ SCREENING ASSAYS FOR ANTIBODY TO HUMAN IMMUNODEFICIENCY VIRUS. The Lancet 1989, 333, (8638).
  2. Debjani, S.; Dipika, R.; Tarun, K.D., Immunofiltration assay for aflatoxin B 1 based on the separation of pre-immune complexes. J Immunol Methods 2013, 392, (1-2).
  3. Zhang, P.; Bao, Y.; Draz, M.S.; Lu, H.; Liu, C.; Han, H., Rapid and quantitative detection of C-reactive protein based on quantum dots and immunofiltration assay. Int J Nanomed 2015, 10, (default).
  4. , A.B.; T., Y.R.; N., A.Y.; S., S.; I., Y.G., Simultaneous Determination of Several Mycotoxins by Rapid Immunofiltration Assay. J Anal Chem.2014, 69, (6).

Results and discussion

Point 4: The advantages of this method in comparison with other methods should be highlighted, including analytical characteristics, reproducibility, specificity, and stability?

Response 4: Thank you very much for your valuable suggestions. In the introduction, we have described the analysis characteristics accordingly. In the results and discussion, we have analyzed reproducibility, specificity, and stability correspondingly.

The supplement in the article is as follows: 

Stability test: Take three batches of test cartridges, named Cartridge A, Cartridge B, and Cartridge C, and test the standard positive serum for brucellosis three times at a time interval of 15 days.

Repeatability detection: Take cassette I, cassette II and cassette III to detect brucellosis standard positive serum, and evaluate the repeatability of RVFT according to the experimental results.

The test results are shown in Table 2. The results show that the RVFT has good stability and good repeatability.

Table 2 Test Results of Stability and Repeatability of Percolation Card

standard brucellosis positive serum

Test card box

1

2

3

Test card box

+

+

+

standard brucellosis positive serum

+

+

+

Test card box

+

+

+

standard brucellosis positive serum

+

+

+

Test card box

+

+

+

standard brucellosis positive serum

+

+

+

Point 5: No data about reversibility are given?

Response 5: Thank you very much for your valuable comments. This article generally does not cover reversibility-related research. If conditions permit, we will improve it step by step.

Point 6: With respect to the preparation method and the effect of temperature:

6.1-There is a significant relationship between the absorbance spectra and the size of the NPs (provide an explanation and insert suitable citations) . 

Response 6.1: Thank you very much for your valuable comments. We have consulted relevant books and literature, provided corresponding explanations, and inserted appropriate citations accordingly.

The supplement in the article is as follows:

UV-visible spectroscopy is a very helpful technique, which can be used to estimate size, concentration and aggregation level of AuNPs. The Localized Surface Plasmon Resonance (LSPR) spectrum depends upon the size and shape of AuNPs. The peak absorbance SPR wavelength increases with particle diameter because the distance between particles decreases due to aggregation [1]. Enhanced SPR peak intensity for smaller nanoparticles (40 nm) compared to larger ones (80 nm) has been reported by Zeng and co-workers [2]. Zuber et al. [1] reported shift in SPR maximum towards longer wavelength with increase in size of AuNPs due to the differences in the frequency of surface plasmon oscillations of the free electrons. The absorbance was also found to be increased with increase in size of NPs due to the enhanced mean free path of the electrons in the larger NPs[3].

  1. Agnieszka, Z.; Malcolm, P.; Erik, S.; Caroline, F.; Benjamin, V.D.H.; David, G.; Andrew, A.; Tanya, M.; Heike, E., Detection of gold nanoparticles with different sizes using absorption and fluorescence based method. Sensors and Actuators: B. Chemical 2016, 227.
  2. Zeng, S.; Xia, Y.; Law, W.C.; Zhang, Y.; Rui, H.; Dinh, X.Q.; Ho, H.P.; Yong, K.T., Size dependence of Au NP-enhanced surface plasmon resonance based on differential phase measurement. Sensors & Actuators B Chemical 2013, 176, (JAN.), 1128-1133.
  3. Yogita, K.; Gurmandeep, K.; Rajesh, K.; Sachin, K.S.; Monica, G.; Rubiya, K.; Ayinkamiye, C.; K., G.; V., V.S.N.R.; Ravichandran, M.; Dipanjoy, G.; Ankit, A.; Rajan, K.; Ankit, K.Y.; Bhupinder, K.; Pankaj, K.S.; Kamal, D.; Omji, P., Gold nanoparticles: New routes across old boundaries. Adv Colloid Interfac 2019, 274, (C).

6.2-It will be better to have a photo of the prepared solutions (to clarify the different colors)   

Response 6.2: Thank you very much for your valuable advice. We have inserted the corresponding picture in the article.

Figure 3. Ultraviolet visible absorption spectra and colloid color diagram of gold nanoparticles prepared at different reaction temperatures

Figure 4. Ultraviolet visible absorption spectra and colloid color diagram of gold nanoparticles prepared by different molar ratios of reactive substances

6.3- The mechanism should be explained based on temperature. Provide more details about the function and choice of the Tween-20 surfactant.

Response 6.3: Thank you very much for your valuable comments. We have inserted in the text the mechanism should be explained based on temperature. At the same time, we have provided more details about the function and choice of the Tween-20 surfactant in the text.

The supplement in the article is as follows:

As we all know, temperature is an important determinant of  the catalytic performance for catalysts [1,2]. Therefore, temperature will affect the quality of gold nanoparticles, and different sizes and shapes of gold nanoparticles will be formed at different temperatures.

In this study, Tween-20 [3], a surfactant, was used to break red blood cells, wet and disperse red blood cells, increase hydrophilicity, and reduce the nonspecific adsorption between antibody and nitrocellulose membrane. 

  1. 1. , H.E.Y.; B., B.; C., L., Electrocatalytic oxidation of glucose at platinum in alkaline medium: on the role of temperature. Journal of Electroanalytical Chemistry and Interfacial Electrochemistry 1988, 246, (2).
  2. 2. Ramón, P.; Kaizheng, Z.; Anna-Lena, K.; Bo, N., Temperature effects on the stability of gold nanoparticles in the presence of a cationic thermoresponsive copolymer. Journal of nanoparticle research: An interdisciplinary forum for nanoscale science and technology 2016, 18, (11).
  3. 3. Jia, M.; Liu, J.; Zhang, J.; Zhang, H., An immunofiltration strip method based on the photothermal effect of gold nanoparticles for the detection of Escherichia coli O157:H7. The Analyst 2019, 144, (2).

Point 7: The validation of this technique should be introduced by comparison with a previously validated method.

Response 7: Thank you very much for your valuable comments. We have supplemented the text with comparisons of RVFT and other methods.

The supplement in the article is as follows:

In the current point-of-care detection technology, lateral flow immunochromatography is a commonly used detection method. Compared with lateral flow immunochromatography, RVFT has three major advantages. First, RVFT shortens the detection time. The detection time of lateral flow immunochromatography is generally 15 to 20 minutes. However, the RVFT detection time is 2-3 minutes, which greatly shortens the detection time. Second, RVFT effectively avoids the hook effect [1], which is an interference associated with excess antigens that do not bind to the conjugated antibody. The last limitation is line interference. It is difficult to use multiple lines in an lateral flow immunochromatography sensor because the front-line reaction interferes with the behind-line reaction.

  1. Fernando, S.A.; Wilson, G.S., Studies of the 'hook' effect in the one-step sandwich immunoassay. J Immunol Methods 1992, 151, (1-2), 47-66.

Thanks again!

We tried our best to improve the manuscript and made some changes in the manuscript.  These changes will not influence the content and framework of the paper. We used the "Track Changes" function to make corresponding Changes, and the Changes are clearly visible.
We appreciate for Editors/Reviewers’ warm work earnestly, and hope that the correction will meet with approval.
Once again, thank you very much for your comments and suggestions.

Sincerely yours

Feng Shi

Round 2

Reviewer 2 Report

Thank the authors for revising your manuscript and responding to all of my comments accordingly. This work may be accepted for publication in Biosensors after the following minor issues are resolved.

1.     Table 1 was supplied by the author to demonstrate the stability of gold nanoparticles. However, these observational results are insufficient; quantifiable data such as DLS size distribution or microscope images are needed.

2.     Clear instructions on how to use this test box to test samples are needed in the Materials and Methods section. The authors solely discuss how to assemble the device and the selection and optimization of the reagents.

Reviewer 3 Report

The authors have satisfactorily addressed most of my concerns. In particular, the authors have significantly streamlined the manuscript by answering all questions and treating all suggestions that are directly related to the reviewers' arguments. With the presented changes, I recommend this manuscript for publication in its current form in the Journal Biosensors.